# Structural phase transition, $s_\pm$-wave pairing, and magnetic stripe order in bilayered superconductor La₃Ni₂O₇ under pressure

Yang Zhang ®[1], Ling-Fang Lin ®[1] ✉, Adriana Moreo ®[1,2], Thomas A. Maier ®[3] ✉ & Elbio Dagotto ®[1,2] ✉

Motivated by the recently discovered high-$T_c$ superconductor La₃Ni₂O₇, we comprehensively study this system using density functional theory and random phase approximation calculations. At low pressures, the Amam phase is stable, containing the $Y^{2-}$ mode distortion from the Fmmm phase, while the Fmmm phase is unstable. Because of small differences in enthalpy and a considerable $Y^{2-}$ mode amplitude, the two phases may coexist in the range between 10.6 and 14 GPa, beyond which the Fmmm phase dominates. In addition, the magnetic stripe-type spin order with wavevector $(\pi, 0)$ was stable at the intermediate region. Pairing is induced in the $s_\pm$-wave channel due to partial nesting between the $\mathbf{M} = (\pi, \pi)$ centered pockets and portions of the Fermi surface centered at the $\mathbf{X} = (\pi, 0)$ and $\mathbf{Y} = (0, \pi)$ points. This resembles results for iron-based superconductors but has a fundamental difference with iron pnictides and selenides. Moreover, our present efforts also suggest La₃Ni₂O₇ is qualitatively different from infinite-layer nickelates and cuprate superconductors.

The recently discovered infinite-layer (IL) nickelate superconductors[1] opened the newest branch of the high-temperature superconductors family[2–6], including materials such as Sr-doped RNiO₂ films (R = Nd or Pr)[1,7] and quintuple-layer nickelate Nd₆Ni₅O₁₂[8]. Similar to the widely discussed high $T_c$ cuprates superconductors[9], the IL nickelates also have a $d^9$ electronic configuration (Ni¹⁺) in the parent phase, as well as a NiO₂ two-dimensional (2D) square layer lattice. However, many theoretical and experimental efforts have revealed that "Ni⁺ is not Cu²⁺"[10], and the fundamental similarities and differences between individual IL nickelate and cuprates have been extensively discussed[6,11–17]. One key difference is that in nickelates two $d$-orbitals ($d_{3z^2-r^2}$ and $d_{x^2-y^2}$) are important, while in cuprates only $d_{x^2-y^2}$ is relevant.

Recently, La₃Ni₂O₇ (LNO) (with the novel $d^{7.5}$ configuration) was reported to be superconducting at high pressure, becoming the first non-IL NiO₂ layered nickelate superconductor[18], with highest $T_c \sim 80$ K. This conclusion was based on measurements of the resistance using a four-terminal device on a sample with an unknown degree of inhomogeneity. Furthermore, they observed a sharp transition and flat stage in resistance, by using KBr as the pressure-transmitting medium, as well as a diamagnetic response in the susceptibility, which it was interpreted as indication of the two prominent properties of superconductivity, zero resistance and Meissner effect[18]. Subsequently, zero resistance has been confirmed by several studies[19–21]. However, the Meissner effect has not been conclusively observed yet. Recently, the potentially "filamentary" nature of the superconducting state has been presented by an experimental group caused by inhomogeneities in the sample[22], providing a tentative explanation for why the Meissner effect has not been observed yet.

LNO displays the reduced Ruddlesden-Popper (RP) perovskite structure. At ambient conditions, LNO has the Amam structure with space group No. 63[23], with an $(a^- \cdot a^- \cdot c^0)$ out-of-phase oxygen octahedral tilting distortion around the [110] axis from the I4/mmm phase[24]. By

[1]Department of Physics and Astronomy, University of Tennessee, Knoxville, TN 37996, USA. [2]Materials Science and Technology Division, Oak Ridge National Laboratory, Oak Ridge, TN 37831, USA. [3]Computational Sciences and Engineering Division, Oak Ridge National Laboratory, Oak Ridge, TN 37831, USA. ✉ e-mail: lflin@utk.edu; maierta@ornl.gov; edagotto@utk.edu

applying pressure of the order of 10 GPa, the $NiO_6$ rotations are suppressed and transform to an Fmmm space group (No. 69)[18]. Then, the Fmmm phase becomes superconducting in a broad pressure range from 14 to 43.5 GPa[18].

Density functional theory (DFT) calculations revealed that the many components of the Fermi surface (FS) are contributed by the Ni orbitals $d_{x^2-y^2}$ and $d_{3z^2-r^2}$. This FS consists of two-electron sheets with mixed $e_g$ orbitals and a hole pocket dominated by the $d_{3z^2-r^2}$ orbital, suggesting a Ni two-orbital minimum model is necessary[25,26]. While completing our work, recent theoretical studies suggested that $s_\pm$-wave pairing superconductivity should dominate, in agreement with our results. This pairing channel is induced by spin fluctuations in the Fmmm phase of LNO[27–31], indicating also the importance of the $d_{3z^2-r^2}$ orbital[25,26,32–34]. Furthermore, the role of the Hund coupling[26,35], electronic correlation effects[32,33,36,37], and the charge and spin instability[25,34,38,39] were also recently discussed.

Interestingly, the Amam to Fmmm phase transition was found around 10 GPa, but the superconductivity was obtained only above 14 GPa[18]. In addition, the values of $T_c$ do not dramatically change in a broad superconducting pressure region in the Fmmm phase of LNO[18]. In this case, several questions naturally arise for LNO: what interesting physics occurs between 10 to 14 GPa? Why is the observed $T_c$ in the superconducting pressure region 14 to 43.5 GPa relatively independent of pressure, as opposed to showing a dome-like dependence? Are the Fmmm structure or pressure itself important for superconductivity? Does the FS topology and $s_\pm$-wave pairing symmetry change in the Fmmm phase under high pressure? What are the main differences between LNO and the previously well-studied bilayered system $Bi_2Sr_2CaCu_2O_8$ (BSCCO) of the cuprate family?

In this work, to answer these questions, we studied in detail the LNO system under pressure, using first-principles DFT and random phase approximation (RPA) calculations. In the low-pressure region (0 to 10.5 GPa), the Amam phase – with the $Y^{2-}$ mode distortion from the Fmmm phase – is stable, while the Fmmm phase is unstable. Due to small differences in enthalpy and a considerable $Y^{2-}$ mode amplitude, between 10.6 and 14 GPa the Amam phase could potentially coexist with the Fmmm phase in the same sample, or leading to sample-dependent behavior, and resulting in a greatly reduced or vanishing $T_c$ in this pressure region. Furthermore, in the range of pressures studied, two sheets ($\alpha$ and $\beta$) with mixed $d_{3z^2-r^2}$ and $d_{x^2-y^2}$ orbitals, and a $\gamma$ pocket made primarily of the $d_{3z^2-r^2}$ orbital contribute to the FS. Compared to ambient pressure, the $\gamma$ pocket is stretched and the $\beta$ sheet is reduced in size. Furthermore, the DFT+$U$ and RPA calculations suggest a stripe spin order instability with wavevector $(\pi, 0)$.

Thus, our results highlight that the main fundamental differences with BSCCO are: (i) in the Ni bilayer with two active Ni orbitals, it is $d_{3z^2-r^2}$ that plays the key role, as compared to $d_{x^2-y^2}$ for BSCCO cuprates (see sketch Fig. 1). (ii) This leads to $s_\pm$-wave pairing for Ni, while it is $d$-wave for Cu. Or, in other words, the inter-layer hybridization being large induces $s^\pm$ in LNO, but when this hybridization is small then $d$-wave dominates as in cuprates. (iii) The FSs of LNO and BSCCO fundamentally differ with regards to the presence of hole pockets at $(\pi, \pi)$ for LNO. These pockets are crucial for the stability of $s_\pm$ pairing.

## Results
### DFT results
Based on the group analysis obtained from the AMPLIMODES software[40,41], the distortion mode from the high symmetry phase (Fmmm) to the low symmetry phase (Amam) is the $Y^{2-}$ (see Fig. 2a). At 0 GPa, the distortion amplitude of the $Y^{2-}$ mode is ~0.7407 Å. As shown in Fig. 2b, this $Y^{2-}$ mode amplitude is gradually reduced under pressure, reaching nearly zero value at 15 GPa (~0.0016 Å). At 0 GPa, the Amam phase has an energy lower by about −21.01 meV/Ni than the Fmmm structure. As shown in Fig. 2b, the difference in enthalpy between the Amam and Fmmm phases also smoothly decreases by

increasing pressure. Interestingly, the Fmmm and Amam phases have very close enthalpies in the pressure range from 9 to 14 GPa, while the $Y^{2-}$ mode distortion still exists with sizeable distortion amplitude in this region. To better understand the structural stability of LNO, we calculated the phonon spectrum of the Fmmm and Amam phases with or without pressure, by using the density functional perturbation theory approach[42–44] analyzed by the PHONONPY software[45,46]. For the Amam phase of LNO, there is no imaginary frequency obtained in the phonon dispersion spectrum from 0 to 15 GPa (see results in the Supplementary Note I), suggesting that the Amam phase is stable in this pressure range.

The phonon dispersion spectrum displays imaginary frequencies appearing at high symmetry points for the Fmmm structure of LNO below 10.5 GPa (see the results of 0 GPa as example, in Fig. 2c). However, the Fmmm phase becomes stable without any imaginary frequency from 10.6 GPa to 50 GPa, the maximum value we studied (see $P = 11$ GPa as an example in Fig. 2d, while the rest of the results can be found in the Supplementary Note I). Between 10.6 and 14 GPa, Fmmm has an enthalpy slightly lower than that of the Amam state (< ~0.3 meV/Ni). This value (~10.6 GPa) is quite close to the experimental observed critical pressure (~10 GPa) for the Amam to Fmmm transition[18].

Recent DFT calculations found that the pocket around $(\pi, \pi)$ vanishes in the Amam phase[18,26], which was also confirmed by angle-resolved photoemission spectroscopy experiments[47]. This pocket induces the $s_\pm$-wave pairing symmetry in the superconducting phase, as discussed below. Due to the small difference in enthalpy and considerable $Y^{2-}$ mode amplitude in this pressure range, the Amam phase could also be obtained experimentally in some portions of the same sample of LNO. Recent experiments also suggest a first-order structural transition from the Amam to Fmmm phases under pressure[18]. Our RPA calculation shows the $s_\pm$-wave pairing superconductivity seems to be unlikely when this $\gamma$ pocket is absent, as discussed in the "Pairing symmetry" section. In this case, the $T_c$ would be greatly reduced or vanish in this pressure region due to the coexistence with the Amam phase.

While finishing the present manuscript, we noted that a very recent experimental effort reported zero resistance below 10 K in some samples above 10 GPa and below 15 GPa[19], supporting our conclusion. This could also qualitatively explain the absence of superconductivity between 10 and 14 GPa in the original high-pressure efforts[18]. At 15 GPa, our DFT results found that the $Y^{2-}$ mode amplitude is almost zero (~0.0016 Å), indicating a pure Fmmm symmetry phase and robust superconductivity above 15 GPa.

### Tight-binding results
Next, we constructed a four-band $e_g$ orbital tight binding (TB) model in a bilayer lattice[48–51] for the Fmmm phase. It is four orbitals because the unit cell contains two Ni's, and each Ni contributes two orbitals. As pressure increases, the values of the hopping matrix elements also increase. As shown in Fig. 3a, the ratio of $t_{11}^z$ ($d_{3z^2-r^2}$ along the interlayer direction) and $t_{22}^{x/y}$ ($d_{x^2-y^2}$ in plane) slightly decreases from 1.325 (0 GPa) to 1.286 (50 GPa), although with some small oscillations. Furthermore, the in-plane $|t_{12}/t_{22}|$ increases from 0.457 (0 GPa) to 0.483 (50 GPa), suggesting an enhanced hybridization between $d_{3z^2-r^2}$ and $d_{x^2-y^2}$ orbitals under pressure (see Fig. 3b). Moreover, the crystal field splitting $\Delta$ also increases under pressure, with small oscillations, as displayed in Fig. 3c. These small oscillations may be caused by a lack of convergence of optimized crystal structures at some pressure values, but do not change the main physical conclusions discussed in our publication.

The TB calculations indicate that the electronic density of the $d_{3z^2-r^2}$ orbital gradually reduces from 1.86 (0 GPa) to 1.78 (50 GPa), as shown in Fig. 3d. Note that the electronic population of both orbitals is not an integer, thus this system is "self-doped". The band structures

indicate that the bandwidth of $e_g$ orbitals increases by about ~23.1%, from 0 (~3.63 eV) to 50 (~4.47 eV) GPa (see Fig. 3e, f). Furthermore, the $e_g$ states of Ni display the orbital-selective spin singlet formation behavior[26], where the energy gap $\Delta E$ between bonding and antibonding states of the $d_{3z^2-r^2}$ orbital increases by about 20% from 0 GPa (~1.20 eV) to 50 GPa (~1.44 eV).

In addition, a van Hove singularity (vHS) near the Fermi level was also found at the $X$ point ($\pi$, 0) in the BZ (see Fig. 3e, f), indicating a possible stripe ($\pi$, 0) order instability. As pressure increases, the vHS shifts away from the Fermi level, leading to reduced magnetic scattering near ($\pi$, 0), as discussed in the following section. We wish to remark that having the vHS at exactly the Fermi energy is not necessary for the stability of the magnetic stripe order. It is sufficient to have the vHS close to that Fermi energy so that the associated wavevector ($\pi$, 0) dominates. The FS consists of two electron sheets ($\alpha$ and $\beta$) with a mixture of $d_{3z^2-r^2}$ and $d_{x^2-y^2}$ orbitals, while the $\gamma$ hole-pocket is made up almost exclusively of the $d_{3z^2-r^2}$ orbital at all pressures we studied

(see $P = 0$ and 50 GPa as examples in Figs. 3g, h, and the Supplementary Note II for other pressures). The $\gamma$ pocket increases in size with pressure, while the size of the $\beta$ pocket decreases at high pressure.

## Stripe order instability

To better understand the tendency towards a possible magnetic instability in LNO under pressure, several possible in-plane magnetic structures of the Ni bilayer spins were considered here: A-AFM with wavevector (0, 0), G-AFM with ($\pi$, $\pi$), and stripe ($\pi$, 0), as shown in Fig. 4a. In all cases, the coupling between the two Ni layers of the bilayer was assumed to be antiferromagnetic (AFM) due to the large interlayer hopping discussed in previous studies[25,26]. Note that the possible in-plane stripe order ($\pi$, 0) can be understood as induced by the strong competition between intraorbital and interorbital dominated hopping mechanisms[52], namely the competition between AFM and ferromagnetic (FM) tendencies that may induce a state with half the bonds AFM and half FM. To discuss the importance of $J$ for the stripe instability, based on the same crystal structures, we used the Liechtenstein formulation within the double-counting item to deal with the onsite Coulomb interactions, where $U$ and $J$ are independent variables[53].

As displayed in Fig. 4b, c, the stripe ($\pi$, 0) magnetic order has the lowest energy among the three magnetic candidates considered in the pressure region that we studied, using robust Hund couplings $J = 0.8$ eV and $J = 1.0$ eV. Furthermore, the energy differences between the stripe and other magnetic configurations decreases as the pressure increases.

Note that the order (from lower to higher energy) is stripe, G-AFM, A-AFM, and finally FM phases, when working at $U = 4$ eV and $J = 0.8$ eV, and at 30 GPa, which is exactly the same as in our previous recent work[26]. However, the energy differences are not the same. The reason is that in our previous work[26] we relaxed the atomic positions and used the experimental lattice constants at 300 K provided by the original discovery publication[18]. However, in our present efforts we optimized the atomic positions and lattice constants at 0 K. This leads to energy differences between various magnetic states, but the relative order of those phases remains the same.

By reducing $J$ to 0.6 eV (see Fig. 4d), the stripe order ($\pi$, 0) has the lowest energy below 25 GPa, while it has a close energy (~0.3 meV/Ni) to the G-AFM phase above 25 GPa, suggesting the important role of $J$ to

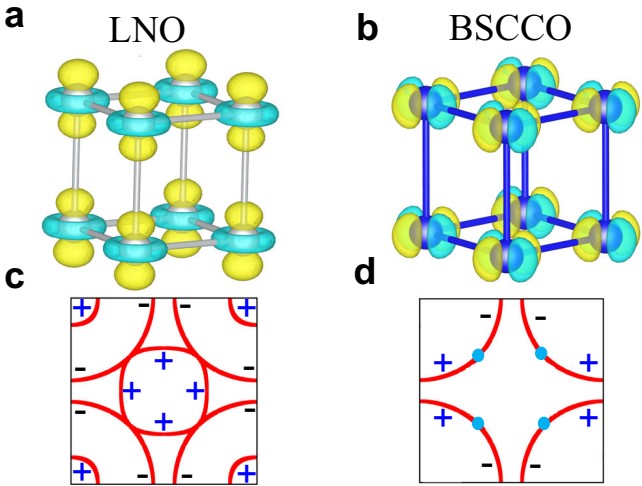

**Fig. 1 | Differences between LNO and BSCCO. a** The dominant orbital $d_{3z^2-r^2}$ of LNO vs. (**b**) the dominant orbital $d_{x^2-y^2}$ of BSCCO. **c, d** Sketches of Fermi surfaces for LNO and BSCCO, including the signs of the superconducting order parameter.

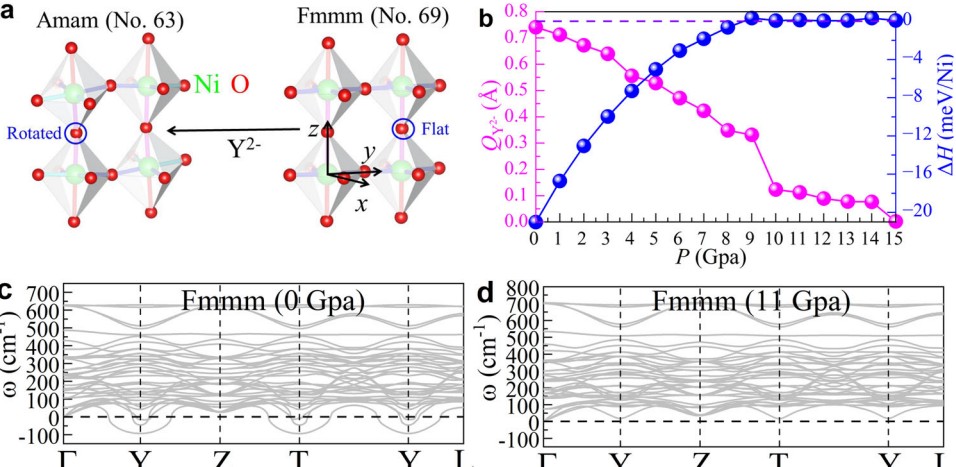

**Fig. 2 | Crystal structures, $Y^{2-}$ distortion amplitude, and phase transition. a** Schematic crystal structure of the bilayer $NiO_6$ octahedron plane of LNO for the Amam (No. 63) and Fmmm (No. 69) phases (green = Ni; red = O), respectively. Different Ni-O bonds are distinguished by different colors. The local $z$-axis is perpendicular to the $NiO_6$ plane towards the top O atom, while the local $x$- or $y$-axis is along the in-plane Ni-O bond directions. All crystal structures were visualized using

the VESTA code[72]. **b** The $Y^{2-}$ distortion amplitude and enthalpy (H = E + PV) between the Amam and Fmmm phases [$\Delta H$ = H(Amam)-H(Fmmm)] vs. pressure. Phonon spectrum of LNO for the **c** Fmmm (No. 69) phase at 0 GPa, and **d** Fmmm (No. 69) phase at 11 GPa, respectively. For the Fmmm phase, the coordinates of the high-symmetry points in the Brillouin zone (BZ) are $\Gamma$ = (0, 0, 0), Y = (0.5, 0, 0.5), Z = (0.5, 0.5, 0), T = (0, 0.5, 0.5), and L = (0.5, 0.5, 0.5).

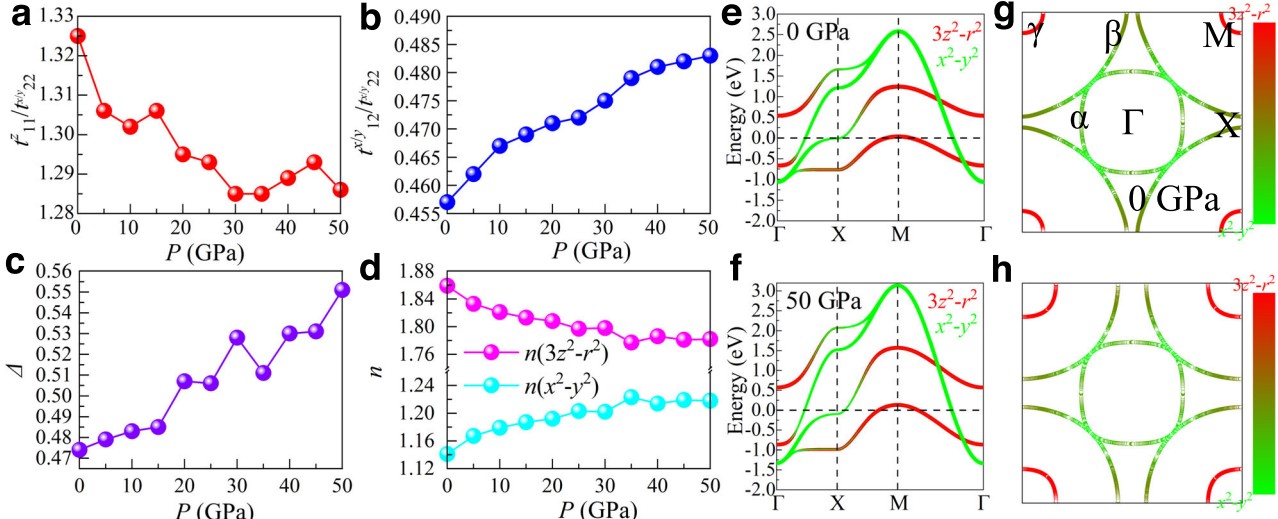

**Fig. 3 | TB results. a** Ratio of hoppings $t_{11}^z/t_{22}^{x/y}$, (**b**) ratio of hoppings $t_{12}^{x/y}/t_{22}^{x/y}$, (**c**) crystal-field splitting $\Delta$, and **d** electronic density $n$, vs. pressure. The $\gamma = 1$ and $\gamma = 2$ orbitals correspond to the $d_{3z^2-r^2}$ and $d_{x^2-y^2}$ orbitals, respectively. **e**, **f** Band structures and **g**-**h** FSs for 0 and 50 GPa, respectively. Here, the four-band $e_g$ orbital TB model was considered with the nearest-neighbor (NN) hoppings in a bilayer lattice for the overall filling $n = 3$ (1.5 electrons per site). The bilayer lattice is shown in Supplementary Note II. The NN hoppings and crystal field $\Delta$s are obtained from the maximally localized Wannier functions[67] by fitting DFT and Wannier bands for different pressures. The hoppings and crystal-field splitting $\Delta$ used at 0 GPa are: $t_{11}^x = t_{11}^y = -0.088$, $t_{12}^x = 0.208$, $t_{12}^y = -0.208$, $t_{22}^x = t_{22}^y = -0.455$, $t_{11}^z = -0.603$, and $\Delta = 0.474$. The hoppings and crystal-field splitting $\Delta$ used at 50 GPa are: $t_{11}^x = t_{11}^y = -0.125$, $t_{12}^x = 0.270$, $t_{12}^y = -0.270$, $t_{22}^x = t_{22}^y = -0.559$, $t_{11}^z = -0.719$, and $\Delta = 0.551$. All the hoppings and crystal-field spliting are in eV units. The coordinates of the high-symmetry points in the BZ are $\Gamma = (0, 0)$, $X = (0.5, 0)$, and $M = (0.5, 0.5)$.

stabilize stripe order. Furthermore, the strongly reduced energy difference indicates the tendency of the strong competition between FM and AFM in the plane direction increasing pressure, suggesting that long-range spin order may not develop in the Fmmm phase under high pressure. Under pressure, the intraorbital hopping of the $e_g$ orbitals increases, enhancing the canonical AFM Heisenberg interaction. Furthermore, the reduced $J$ would also reduce the FM coupling caused by the interorbital hopping between half-filled and empty orbitals via Hund's coupling $J$[26,52]. If continuing to reduce $J$, then the AFM Heisenberg interaction induced by the intraorbital hopping will eventually win, leading to G-type AFM order (see the results for $J = 0.4$ eV in Supplementary Note III).

Considering quantum fluctuations, the system may not develop long-range order due to the in-plane AFM and FM competition in portions of the vast parameter space involving hoppings, Hund coupling $J$, and Hubbard interaction $U$. This competition deserves further many-body model studies. In addition, the calculated magnetic moment of the magnetic stripe phase also decreases under pressure (Fig. 4e) because increasing the hoppings (namely, increasing the bandwidth $W$) "effectively" reduces the electronic correlation via $U/W$. Finally, note that the stripe order $(\pi, 0)$ is degenerate with stripe order $(0, \pi)$. Thus, there could occur an Ising spontaneous symmetry breaking upon cooling before long-range order is reached. In the context of the study of iron-based superconductors, "nematicity" was extensively discussed based on having in-plane stripe $(\pi, 0)$ spin order[54,55]. The essence of this phenomenon is that the stripe states with wavevectors $k_1 = (\pi, 0)$ and $k_2 = (\pi, 0)$ should be degenerate by symmetry and, thus, at high temperature their spin structure factors $S(k)$ should be equal. Then, upon cooling two transitions could potentially exist. At the first one, say $T_{nem}$, the spin structure factor of these two wave vectors becomes different, signalling a dominance of one stripe over the other. This is the "nematic" state where rotational invariance by $90^o$ degrees is spontaneously broken, but there is yet no long range order. At a lower temperature, the true Néel temperature, long-range order is finally established. Consequently, our theoretical results for LNO indicate the possible existence of "nematicity" in LNO as well, as it occurs in iron-based superconductors[54,55]. However, this issue certainly merits further investigation and detailed discussion that is left for future studies.

To assess the DFT extracted TB models for their magnetic and superconducting behavior, we have performed multi-orbital RPA calculations (see Methods section) for the Fmmm phase of LNO. Figure 5 shows the static RPA enhanced spin susceptibility $\chi'(\mathbf{q}, \omega = 0)$ for $q_z = \pi$ and $q_x$, $q_y$ along a high-symmetry path in the Brillouin zone. At 0 GPa, $\chi'(\mathbf{q}, \omega = 0)$ has a strong peak near the stripe wavevector q $= (\pi, 0)$. A closer inspection of the contributions to the spin susceptibility shows that the dominant scattering process giving rise to this peak comes from intra-orbital $d_{3z^2-r^2}$ scattering between the $(0, \pi)$ region on the $\beta$ electron sheet and the $\gamma$ hole pocket at $(\pi, \pi)$ (see the results in Supplementary Note IV). In addition, we also wish to remark that this strong increase of the magnetic scattering at this wavevector is much enhanced by the vHs that happens at $X$ point in our TB calculation. The peak in the spin susceptibility of Fig. 5 at ambient pressure will be much reduced if the vHS shifts further down from the Fermi level. Thus, the vHS is crucial for the sharp peak features in the spin susceptibility. Because the Fmmm phase is not stable at 0 GPa, here we will not elaborate further about the strong influence of the vHS on the susceptibility.

For larger pressures, this saddle-point moves away from the Fermi level (see Fig. 3f), and, as a consequence, the magnetic scattering near $(\pi, 0)$ is reduced, as shown in Fig. 5. This is also in agreement with the decreasing energy differences between stripe $(\pi, 0)$ and other magnetic states under pressures obtained from DFT+$U$ calculations (see Fig. 4). Furthermore, the huge reduction of magnetic scattering under pressure also suggests the long-range spin stripe order may not be stable at high pressure, which may explain the absence of long-range order in the Fmmm phase under pressure. However, the short-range order is still possible.

## Pairing symmetry

In the RPA approach, the spin (and also charge) susceptibilities enter directly in the pairing interaction for the states on the FS. Figure 6 displays the leading pairing symmetry $g_\alpha(\mathbf{k})$ obtained from solving the eigenvalue problem in Eq. (3) (Methods section) for the RPA pairing

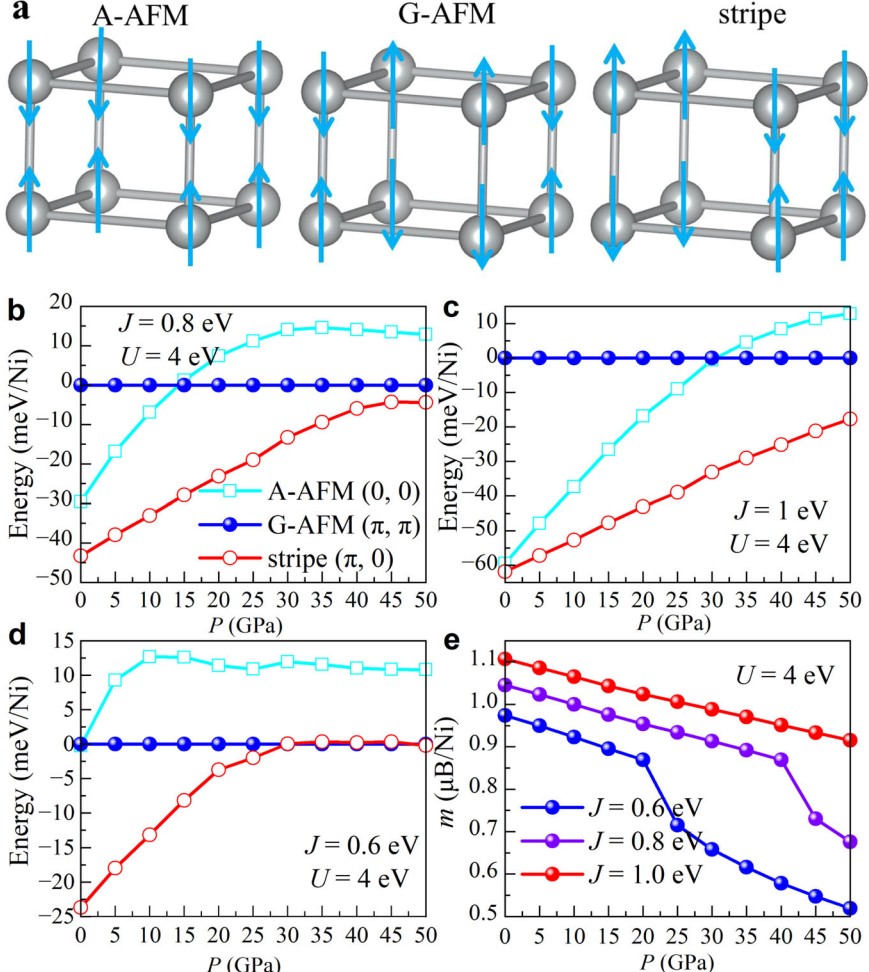

**Fig. 4 | Magnetic results. a** Sketch of three possible in-plane configurations (spins denoted by arrows) in a bilayer Ni lattice: A-AFM with wavevector (0, 0), G-AFM with $(\pi, \pi)$, and stripe $(\pi, 0)$ using the optimized crystal structures at different pressures. Here, the coupling between the two Ni layers is assumed AFM. The DFT+$U$+$J$ calculated energies for (**b**) $J = 0.8$ eV, (**c**) $J = 1$ eV, and (**d**) $J = 0.6$ eV of different magnetic configurations vs different values of pressure for the Fmmm phase of LNO, all at $U = 4$ eV, respectively, taking the G-AFM state as zero of reference. **e** The calculated

magnetic moment of the stripe $(\pi, 0)$ magnetic order for different values of $J$, at $U = 4$ eV. To better understand the Hund coupling role, $J$ was changed from 0.6 to 1.0 eV by fixing $U = 4$ eV, by using the Liechtenstein formulation within the double-counting item[53]. The magnetic coupling between layers was considered to be AFM in all cases. The differences in total energy and enthalpy between different magnetic configurations are the same due to the same crystal structures at different pressures.

interaction for the model at (a) 0 GPa and (b) 25 GPa. In both cases, the leading superconducting gap has an $s^{\pm}$ structure, where the gap switches sign between the $\alpha$ and $\beta$ sheets, and between the $\beta$ sheet and $\gamma$ pocket. As for the spin susceptibility, and as one would expect for spin-fluctuation mediated pairing, a detailed analysis of the different contributions to the $s^{\pm}$ pairing strength reveals that this gap structure is driven by intra-orbital $(\pi, 0)$ scattering between the $(0, \pi)$ region of the $\beta$-sheet with significant $d_{3z^2-r^2}$ character of the Bloch states, to the $d_{3z^2-r^2}$ $\gamma$-pocket at $(\pi, \pi)$. Moreover, the gap amplitude on the $\alpha$ sheet grows relative to that on the $\beta$ sheet and $\gamma$ pocket with increasing pressure. Also, independent of pressure, the gap on the $\beta$ pocket has strong momentum dependence, becoming very small near the zone diagonal where it has accidental nodes. We reserve a detailed analysis of the factors leading to this momentum dependence for a future study.

In addition, we show in Fig. 7 the pressure dependence of the pairing strength $\lambda$ of the leading $s^{\pm}$ gap. The $s^{\pm}$ gap remains the leading instability over the subleading $d_{x^2-y^2}$ gap over the entire pressure range we considered. With increasing pressure, both pairing strengths monotonically decrease. In the RPA approach we use, changes in the pairing strength $\lambda$ translate to changes in the superconducting

transition temperature $T_c$ through a Bardeen-Cooper-Schrieffer (BCS) like equation, $T_c = \omega_0 e^{-1/\lambda}$, where $\omega_0$ is a cut-off frequency that is determined from the spin-fluctuation spectrum. We wish to remark that the quite drastic increase of the pairing strength in the $s^{\pm}$ channel as we reach ambient pressure is much enhanced by the vHS. But even if the vHS would not be exactly at the Fermi energy, to the extent that it is simply close to the Fermi energy would be sufficient for dominance of the $s^{\pm}$ channel. Furthermore, our RPA shows the pairing strength in the $s^{\pm}$ channel is considerable in the pressure region we studied, indicating a broad superconducting region, which can explain qualitatively that superconductivity was found in a broad pressure region in the experiment at the region they studied (14–43.5 GPa)[18].

Moreover, very recent angle-resolved photoemission spectroscopy experiment reveals that the hole $\gamma$-pocket made of $d_{3z^2-r^2}$ was absent in the Amam phase at ambient pressure[47], while this pocket was found in the DFT studies in the high-pressure Fmmm phase[18,25,26]. To understand the importance of this hole $\gamma$-pocket made of $d_{3z^2-r^2}$, here, we also performed additional calculations for a model with artificially large crystal-field splitting $\Delta = 0.6$ eV at 0 GPa, for which the hole band sinks below the Fermi level and the $\gamma$-pocket disappears (see the results in Supplementary Note V). For this case, $\lambda_{s^{\pm}}$ is suppressed significantly

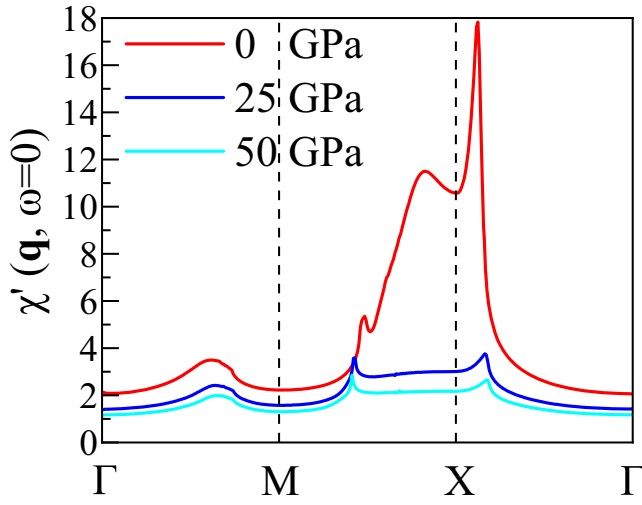

**Fig. 5 | RPA magnetic susceptibility.** The RPA calculated static spin susceptibility $\chi'(\mathbf{q}, \omega = 0)$ versus $q_x, q_y$ for $q_z = \pi$ for the two-orbital bilayer TB model for three different pressures. At 0 GPa, $\chi'(\mathbf{q}, \omega = 0)$ shows a strong peak at q = $(\pi, 0)$ (and symmetry related wavevectors), which is supressed at higher pressure. Here we used $U = 0.8, U' = 0.4, J = J' = 0.2$ in units of eV. All hoppings and crystal fields are for the Fmmm phase.

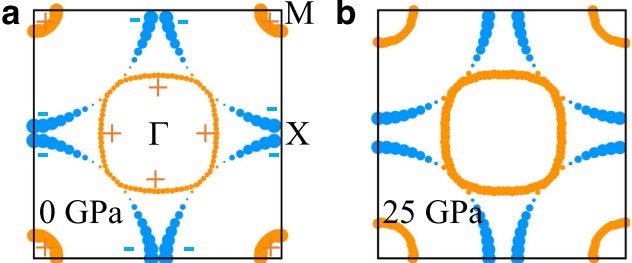

**Fig. 6 | Leading RPA gap structure.** The RPA calculated leading superconducting gap structure $g_\alpha(\mathbf{k})$ for momenta k on the FS for (**a**) 0 GPa and (**b**) 25 GPa has $s^\pm$ symmetry, where the gap changes sign between the $\alpha$ and $\beta$ FS sheets, and also between the $\beta$ sheet and $\gamma$ pocket. The sign of the gap is indicated by color (orange = positive, blue = negative), and gap amplitude by thickness. With increasing pressure, the gap amplitude on the $\alpha$ sheet grows relative to that on the $\beta$ sheet and $\gamma$ pocket. Independent of pressure, the gap on the $\beta$ sheet is very small and has nodal points near the zone diagonal. Hoppings and crystal fields are for the Fmmm phase.

from 1.55 (at $\Delta = 0.474$ eV) to 0.040, and $\lambda_{d_{x^2-y^2}}$ becomes the leading solution, albeit with a similarly small $\lambda_{d_{x^2-y^2}} = 0.045$. Consistent with the discussion above, this provides further evidence of the importance of the $(\pi, \pi)$ $\gamma$ hole pocket in mediating superconductivity in this system. This could explain the absence of superconductivity in the low-pressure Amam phase of LNO[18], where the $\gamma$ pocket around $(\pi, \pi)$ vanishes, indicating the importance of the Fmmm phase for the superconductivity in LNO system.

It may occur that small variations in the Hubbard $U$ may lead to qualitatively different results. For this reason, we varied $U$ in an allowed range before a spin-density-wave state starts dominating at 25 GPa in the RPA context. This requirement establishes $U = 1.05$ as the upper limit that can be studied within our RPA formalism. In Table 1, we provide the values of $\lambda$ for the $s_\pm$ and $d$-wave channels with increasing $U$ in that allowed range. The results show that $\lambda$ increases smoothly with increasing $U$ and our study produces a dominant $s_\pm$ pairing state in the entire range analyzed[56], suggesting that our results are stable under small variations of the Hubbard strength.

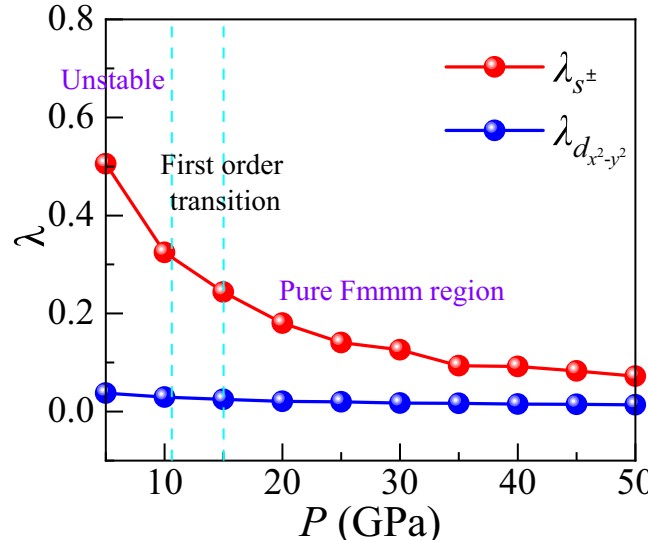

**Fig. 7 | Pressure dependence of leading pairing strength.** The RPA calculated pairing strength $\lambda$ for the leading $s^\pm$ and subleading $d_{x^2-y^2}$ instabilities versus pressure for the pressure dependent bilayer model with $U = 0.8, U' = 0.4, J = J' = 0.2$ in units of eV. The $s^\pm$ instability is leading over the full pressure range and its pairing strength increases monotonically with decreasing pressure. All hoppings and crystal fields are for the Fmmm phase.

**Table 1 | Pairing results in the $s_\pm$ and $d$-wave channels corresponding to the range of $U$ allowed by the RPA formalism before a spin-density-wave state starts dominating**

| Table I–Pairing strength $\lambda$ vs $U$ | | |
|---|---|---|
| $U$ | $\lambda_s$ | $\lambda_d$ |
| 0.80 | 0.141 | 0.020 |
| 0.82 | 0.162 | 0.023 |
| 0.84 | 0.186 | 0.026 |
| 0.86 | 0.215 | 0.030 |
| 0.88 | 0.249 | 0.035 |
| 0.90 | 0.290 | 0.041 |
| 0.92 | 0.339 | 0.049 |
| 0.94 | 0.400 | 0.060 |
| 1.00 | 0.691 | 0.125 |
| 1.05 | 1.264 | 0.330 |

In the entire range studied, the $s_\pm$ state is the prevailing superconducting channel.

Note that above $U = 1.05$, where magnetic order develops, in principle pairing could still occur. To address this matter, the RPA formalism must be generalized by carrying out the resummation of bubble diagrams now using, e.g., dressed propagators corresponding to the dominant magnetic order. This task is demanding and results will be presented in future work.

## Charge order instability
In the experimental studies, they excluded the possibility of the presence of charge-density-wave order at low temperatures under pressures from 14.0 to 43.5 GPa, based on the resistance measurements[18]. Here, we will also briefly discuss charge order (CO) instabilities in the Fmmm phase under pressure, by carrying out DFT calculations. The checkerboard CO (G-type) configuration was considered, with both AFM coupling in-plane and between the two Ni layers. This checkerboard charge order was proposed in another bilayer nickelate La$_3$Ni$_2$O$_6$[57]. Here, we studied two specific pressure values (15 GPa and

30 GPa) in the Fmmm phase, with the two values in the "super-conducting" region of the phase diagram in the experiments[18,21].

First, we used the specific values $U = 4$ eV and $J = 0.6$ eV, very similar to those obtained from constrained density functional calculations ($U \sim 3.8$ eV and $J \sim 0.61$ eV)[33], as shown in Table 2. Without lattice relaxations, in our study at 15 GPa, we observed a strong charge disproportionation in this charge order state with values 0.933 and 0.377 $\mu_B$/Ni for two different Ni sites, but this state has a higher energy of about ( ~ 8.11 meV/Ni) than the magnetic stripe ground state. Moreover, we do not obtain any obvious charge disproportionation at 30 GPa, as displayed in Table 2. In addition, we also considered the possibility of structural distortions for different magnetic configurations, namely lattice relaxations, and this enhances the charge disproportionation in the checkerboard CO state, but our conclusions above did not change. Increasing $J$ to 0.8 eV, by introducing the structural distortions, the charge disproportionation increases to 1.236/0.364 and 0.953/0.395 $\mu_B$/Ni for 15 and 30 GPa, respectively, in this charge order state. Even though the charge disproportionation is enhanced in this charge order state with higher $J$, this state still has higher enthalpy ( ~ 17.47 and 20.89 meV/Ni) than the magnetic stripe state at 15 and 30 GPa. Hence, from our DFT perspective, the charge density wave state is not stable in the Fmmm phase under pressure, at least at low temperatures. This is in agreement with the experimental observations.

### Comparison with IL nickelates and cuprates superconductors

The discovery of high $T_c \sim 80$ K in LNO naturally reminds us of the recently discussed IL nickelates as well as the previous widely studied cuprates superconductors, and show some significant differences. The experimentally observed phase diagram of the bilayer LNO is significantly different from the previously reported phase diagram of IL nickelates and cuprates superconductors, where no obvious sharp and narrow superconducting dome was obtained[18–21]. On the contrary, a very broad pressure superconducting region was observed in LNO. This suggests that LNO system is quite unique, compared with IL nickelates and cuprates superconductors.

Based on our results described here, the strong inter-layer coupling caused by the $d_{3z^2-r^2}$ orbital leads to possible $s_\pm$-wave pairing superconductivity in LNO in a broad pressure region. However, the inter-layer coupling in IL nickelates is weak[2,5]. These combined results suggest that the $d_{3z^2-r^2}$ orbital plays a quite different role in LNO and in the previously discovered IL nickelates. In addition, a robust inter-orbital hopping between $e_g$ orbitals was found via DFT in LNO, while this hopping is nearly zero in the IL nickelates, leading to in-plane stripe vs G-AFM spin order in those two systems, as discussed above[6,26]. These two characteristics are the main differences between the two systems.

In the cuprates, superconductivity is believed to be driven by the in-plane AFM fluctuations of the $d_{x^2-y^2}$ orbital, resulting in $d$-wave superconductivity[9]. In addition, the inter-layer coupling is weak in the cuprates. Furthermore, oxygen also plays a different role in the bilayer LNO and cuprates superconductors. LNO has a large charge-transfer gap from oxygen $p$ to Ni's $3d$ orbitals[26], resulting in being more close to a Mott-Hubbard system. But this charge-transfer gap is quite smaller in the cuprate superconductors, leading to a charge-transfer system[6]. Moreover, in the context of the study of cuprates, it is well known that $T_c$ increases substantially going from 1 layer to 2 layers, and then slightly increases further in the case of 3 layers[58]. However, the RP Ni-oxide layered materials are not simply following the same trend as the cuprates: the superconductivity was not yet observed in single-layer at the pressure region studied[21] but LNO is already superconducting in this region. In addition, the superconductivity was also not found in a earlier study[21] but signatures of superconductivity were reported in trilayer RP nickelate under pressure[59]. Interestingly, the inter-layer coupling is also weak in the single-layer, suggesting the importance of inter-layer coupling for the superconductivity in Ni-oxide layered materials. All these results combined strongly indicate that LNO is qualitatively different from both IL nickelates and cuprates superconductors.

In addition, for completeness we also considered a single $d_{x^2-y^2}$-orbital model for BSCCO at a filling of $n = 0.85$, based on the hopping obtained from previous work[60]. To find the same $\lambda$ as obtained for LNO at 25 GPa, we need a ~ 25% larger $U$ for BSCCO, which seems reasonable because cuprates are widely believed to have stronger electronic correlations than nickelates. However, it should be noted that a similar $\lambda$ does not necessarily translate to the same $T_c$, because $T_c = \omega_0 e^{-1/\lambda}$. Then, the cut-off frequency prefactor $\omega_0$ also plays a role, and $\omega_0$ is different in those two systems. To provide a more quantitative comparison in $T_c$ between LNO and a typical cuprate, many additional complex effects must be incorporated, more orbitals are need in the model calculations, as well as a different cut-off frequency prefactor $\omega_0$ if the RPA formalism is still being used. But at least the qualitative trends appear to be correctly reproduced: cuprates require a larger $U$ than nickelates.

### Discussion

The recently discovered bilayer nickelate superconductor LNO has opened a new platform for the study of the origin of unconventional superconductivity, unveiling several challenging results that theory needs to explain. Combining first-principles DFT and many-body RPA methods, here we comprehensively studied the LNO system under pressure from $P = 0$ GPa to $P = 50$ GPa. Based on group analysis, the distortion $Y_2^-$ mode induces the structural transition from Fmmm (No. 69) to Amam (No. 63). At 0 GPa, the Amam phase has lower energy ( ~ −21.01 meV/Ni) than the Fmmm phase due to a large

**Table 2 | Energy differences (meV/Ni) and calculated magnetic moment ($\mu_B$/Ni) for the various input spin configurations by using the same lattice structure**

| Table II–Charge order instability | | |
| --- | --- | --- |
| **15 GPa ($J = 0.6$ eV)** | | |
| **Magnetism** | **Energy(Enthalpy** | **Magnetic moment** |
| Stripe | 0(0) | 0.895(0.933) |
| G-AFM | 8.17(16.10) | 0.630(0.650) |
| A-AFM | 20.77(30.10) | 0.598(0.650) |
| CO | 8.11(13.89) | 0.933/0.377(0.968/0.335) |
| **30 GPa ($J = 0.6$ eV)** | | |
| **Magnetism** | **Energy(Enthalpy** | **Magnetic moment** |
| Stripe | 0(0) | 0.658(0.702) |
| G-AFM | −0.03(3.82) | 0.560(0.574) |
| A-AFM | 11.91(16.58) | 0.510(0.528) |
| CO | −0.03(3.80) | 0.561/0.560(0.600/0.549) |
| **15 GPa ($J = 0.8$ eV)** | | |
| **Magnetism** | **Energy(Enthalpy** | **Magnetic moment** |
| Stripe | 0(0) | 0.976(1.009) |
| G-AFM | 27.90(37.63) | 0.725(0.746) |
| A-AFM | 29.18(36.01) | 1.027(1.067) |
| CO | 24.31(17.47) | 1.041/0.368(1.236/0.364) |
| **30 GPa ($J = 0.8$ eV)** | | |
| **Magnetism** | **Energy(Enthalpy** | **Magnetic moment** |
| Stripe | 0(0) | 0.913(0.944) |
| G-AFM | 13.24(22.30) | 0.651(0.667) |
| A-AFM | 27.30(35.98) | 0.726(0.780) |
| CO | 13.71(20.89) | 0.795/0.514(0.953/0.395) |

Here, the stripe configuration is taken as the reference of energy. In addition, we also considered the possibility of structural distortions, namely lattice relaxations, and all these results are shown in parenthesis.

distortion amplitude of $Y^{2-}$ mode ( ~ 0.7407 Å). By introducing pressure, the $Y^{2-}$ mode amplitude is gradually reduced, reaching nearly zero value at 15 GPa ( ~ 0.0016 Å), while the enthalphy difference [$\Delta H$ = H(Amam)-H(Fmmm)] also decreases. Furthermore, there is no imaginary frequency obtained for the Amam structure from 0 to 15 GPa. The Fmmm phase eventually becomes stable at ~ 10.6 GPa, which is quite close to the experimentally observed critical pressure ( ~ 10 GPa) of the Fmmm structure of LNO.

In the pressure range from 10.6 to 14 GPa, we found that the enthalpy differences between the Amam and Fmmm phases are quite small ( ~ 0.3 meV/Ni), while the $Y^{2-}$ mode amplitude is still considerable ( ~ 0.1 Å). Furthermore, both the Amam and Fmmm phases are stable in this pressure region. In this case, the Amam phase could coexist with the Fmmm phase in samples of LNO, suggesting a first-order pressure-induced structural phase transition from Amam to Fmmm. Due to the existence of the Amam phase in the Fmmm structure of the LNO sample, as an overall effect the superconducting $T_c$ would be gradually reduced or fully vanish in the intermediate pressure region, leading to potentially sample-dependent issues, supporting recent experiments where zero resistance was found below 10 K in some samples above 10 GPa and below 15 GPa[19]. This could qualitatively explain the absence of superconductivity in LNO for the Fmmm structure in the pressure region from 10 to 14 GPa[18]. More detailed studies are needed to confirm our results and obtain direct experimental evidence for the coexistence of the Amam and Fmmm phases in this pressure region.

Furthermore, a vHs near the Fermi level was found at the $X$ ($\pi$, 0) point in the BZ in our TB band structure, indicating a possible stripe ($\pi$, 0) order instability. Our DFT+$U$+$J$ and RPA results both indicate that the magnetic stripe phase with wavevector ($\pi$, 0) [degenerate with (0, $\pi$)] should dominate once Hubbard and Hund correlation effects are taken into account, at least in the intermediate range of Hubbard-Hund couplings. Moreover, due to the particular shape of the FS, and its orbital composition, the $s^{\pm}$ pairing channel should be the pairing symmetry of LNO. The subleading superconducting instability was the $d_{x^2-y^2}$ state. Furthermore, we also found that the $d$-wave pairing channel has energy close to the $s$-wave pairing channel. Thus, in experiments, depending on the specific chemical formula and pressure and sample quality, experimentalists may see $s_{\pm}$-wave or $d$-wave superconductivity tendencies in this bilayer family, or even coexisting signals, which also deserves further experimental and theoretical studies.

The significant suppression of the $s^{\pm}$ pairing strength we find in calculations without the $\gamma$ pocket provide strong evidence of the importance of the ($\pi$, $\pi$) $\gamma$ hole pocket in mediating superconductivity in this system. This could also explain why superconductivity is only observed in the high-pressure Fmmm phase since the $\gamma$ hole pocket around ($\pi$, $\pi$) vanishes in the Amam phase, suggesting the importance of the Fmmm phase (stable $\gamma$ pocket) for superconductivity in the LNO system. In addition, from our DFT perspective, the charge density wave state seems to be unlikely in the Fmmm phase under pressure at least at low temperatures. In addition, our efforts in this work also indicate the bilayer nickelate LNO superconductor is unique, because it is qualitatively different from previously discussed IL nickelates and cuprate superconductors. However, to provide a good quantitative comparison in $T_c$ between LNO and cuprates, as well as IL nickelates, many complicated effects and more orbitals need to be considered in the further model calculations.

## Methods
### DFT method
In this work, first-principles DFT calculations were implemented based on the Vienna ab initio simulation package (VASP) code, by using the projector augmented wave (PAW) method[61–63]. The electronic correlations were considered by the generalized gradient approximation

and the Perdew-Burke-Ernzerhof exchange potential[64]. The plane-wave cutoff energy was set as 550 eV and a $k$-point grid $12 \times 12 \times 3$ was adopted for the conventional structure of LNO of both the Amam and Fmmm phases. Note we also tested that this $k$-point mesh produces converged energies. Moreover, the lattice constants and atomic positions were fully relaxed until the Hellman-Feynman force on each atom was smaller than 0.01 eV/Å. Moreover, the onsite Coulomb interactions were considered via the Dudarev formulation[65]. Here, we used the value $U_{eff}$ = 4 eV in the relaxation of crystal structures under pressure, following recent DFT studies of LNO[18,33,38]. Our optimized lattice parameters are $a$ = 5.434 Å, $b$ = 5.367 Å, and $c$ = 20.670 Å for the Amam phase at ambient conditions, which is in good agreement with the experimental values, such as in the neutron data ($a$ = 5.448 Å, $b$ = 5.393 Å, and $c$ = 20.518 Å)[23] and in recent X-ray diffraction ($a$ = 5.438 Å, $b$ = 5.400, and $c$ = 20.455 Å)[66]. We also notice that a recent linear response study suggests ~ 6 eV for LNO[18]. However, our optimized lattice parameters of the Amam phase without pressure for $U_{eff}$ = 6 eV are $a$ = 5.366 Å, $b$ = 5.299 Å, $c$ = 21.429 Å, in good agreement with experiments. While the calculation of structural parameters does not provide sufficient basis for our specific choice of $U_{eff}$, considering together all the results mentioned above, it is clear that the choice of $U_{eff}$ = 4 eV is better than $U_{eff}$ = 6 eV.

We calculated the phonon spectra of the Amam and Fmmm phases for different pressures by using the density functional perturbation theory approach[42–44], analyzed by the PHONONPY software in the primitive unit cell[45,46]. Furthermore, the onsite Coulomb interactions were considered via the Dudarev formulation[65] with $U_{eff}$ = 4 eV. Here, we considered a pressure grid with an interval of 1 Gpa from 0 to 15 Gpa for the Amam and Fmmm phases, but the interval changed to 0.1 Gpa near the critical pressures. In addition, a pressure grid with an interval of 5 Gpa was used for the Fmmm phase from 15 to 50 Gpa. To avoid repeating too many displays, we only show the phonon spectrum of a few key values of pressures in both the main text and in the supplemental materials (see these results in Supplementary Note I). We chose the conventional cell structures, corresponding to the $\sqrt{2} \times \sqrt{2} \times 1$ supercell of the undistorted parent $I4/mmm$ conventional cell, in order to study the dynamic stability. This is enough to obtain possible unstable modes for the 327-type RP perovskite system. In addition to the standard DFT calculation discussed thus far, the maximally localized Wannier functions (MLWFs) method was employed to fit Ni's $e_g$ bands and obtain the hoppings and crystal-field splitting for model calculations, by using the WANNIER90 package[67].

To better understand the possible magnetic instabilities and the importance of the Hund coupling $J$ in the LNO system under pressure, we also used the DFT plus $U$ and $J$ within the Liechtenstein formulation using the double-counting item[53], where $U$ was fixed to be 4 eV and $J$ was changed from 0.4 to 1 eV. In addition, we also show the band structures of the magnetic stripe phase from $J$ = 1 eV to $J$ = 0.4 eV at 25 GPa in the Supplementary Note III.

### TB method
In our TB model, a four-band bilayer lattice with filling $n$ = 3 was used, corresponding to 1.5 electrons per site, where the kinetic hopping component is:

$$H_k = \sum_{\substack{i\sigma \\ \vec{\alpha}\gamma\gamma'}} t^{\vec{\alpha}}_{\gamma\gamma'} \left( c^{\dagger}_{i\sigma\gamma} c_{i+\vec{\alpha}\,\sigma\gamma'} + H.c. \right) + \sum_{i\gamma\sigma} \Delta_{\gamma} n_{i\gamma\sigma}. \tag{1}$$

The first term represents the hopping of an electron from orbital $\gamma$ at site $i$ to orbital $\gamma'$ at the nearest-neighbor (NN) site $i + \vec{\alpha}$. $c^{\dagger}_{i\sigma\gamma}(c_{i\sigma\gamma})$ is the standard creation (annihilation) operator, $\gamma$ and $\gamma'$ represent the different orbitals, and $\sigma$ is the $z$-axis spin projection. $\Delta_{\gamma}$ represents the

crystal-field splitting of each orbital $\gamma$. The unit vectors $\vec{\alpha}$ are along the three directions.

The NN hopping matrix of different pressures was obtained from MLWFs. The detailed values can be found in Supplementary Note II. The Fermi energy is obtained by integrating the density of states for all $\omega$ to reach the number of electrons $n = 3$. Based on the obtained Fermi energy, a $4000 \times 4000$ k-mesh was used to calculate the Fermi surface. The main characters of those Fermi surfaces, namely hole pocket $\gamma$ and two electron sheets ($\alpha$ and $\beta$) are qualitatively in agreement with the present DFT and TB studies[25,29,32,32,39]. However, the vHs just "happen" at the $X$ point in our TB band with nearest-neighbor hopping for 0 GPa, but this singularity would be softened by adding additional hoppings beyond nearest-neighbors, or by other means such as broadening by disorder. A small shift of the flat band at $X$ from the Fermi level cannot change the physics much by mere continuity.

## RPA method

The RPA method we used to assess the bilayer TB models for their magnetic and superconducting behavior is based on a perturbative weak-coupling expansion in the Coulomb interaction. It has been shown in many studies to capture the essence of the physics (e.g. ref. [68]). The full Hamiltonian for the bilayer Hubbard model discussed here, includes the kinetic energy and interaction terms, and it is written as $H = H_k + H_{int}$.

The electronic interaction portion of the Hamiltonian includes the standard same-orbital Hubbard repulsion $U$, the electronic repulsion $U'$ between electrons at different orbitals, the Hund's coupling $J$, and the on-site inter-orbital electron-pair hopping terms ($J'$). Formally, it is given by:

$$H_{int} = U \sum_{i\gamma} n_{i\uparrow\gamma} n_{i\downarrow\gamma} + \left(U' - \frac{J}{2}\right) \sum_{\substack{i \\ \gamma<\gamma'}} n_{i\gamma} n_{i\gamma'} \\ - 2J \sum_{\substack{i \\ \gamma<\gamma'}} \mathbf{S}_{i,\gamma} \cdot \mathbf{S}_{i,\gamma'} + J \sum_{\substack{i \\ \gamma<\gamma'}} (P_{i\gamma}^\dagger P_{i\gamma'} + H.c.),$$

(2)

where the standard relation $U' = U - 2J$ and $J' = J$ are assumed, and $P_{i\gamma} = c_{i\downarrow\gamma} c_{i\uparrow\gamma}$. Thus, there are only two free parameters.

In the multi-orbital RPA approach[69–71], the RPA enhanced spin susceptibility shown in Fig. 5 is obtained from the bare susceptibility (Lindhart function) $\chi_0(\mathbf{q})$ as $\chi(\mathbf{q}) = \chi_0(\mathbf{q})[1 - \mathcal{U}\chi_0(\mathbf{q})]^{-1}$. Here, $\chi_0(\mathbf{q})$ is an orbital-dependent susceptibility tensor and $\mathcal{U}$ is a tensor that contains the intra-orbital $U$ and inter-orbital $U'$ density-density interactions, the Hund's rule coupling $J$, and the pair-hopping $J'$ term. The pairing strength $\lambda_\alpha$ for channel $\alpha$ shown in Fig. 7 and the corresponding gap structure $g_\alpha(\mathbf{k})$ shown in Fig. 6 are obtained from solving an eigenvalue problem of the form

$$\int_{FS} d\mathbf{k}' \, \Gamma(\mathbf{k} - \mathbf{k}') g_\alpha(\mathbf{k}') = \lambda_\alpha g_\alpha(\mathbf{k}),$$

(3)

where the momenta $\mathbf{k}$ and $\mathbf{k}'$ are on the FS and $\Gamma(\mathbf{k} - \mathbf{k}')$ contains the irreducible particle-particle vertex. In the RPA approximation, the dominant term entering $\Gamma(\mathbf{k} - \mathbf{k}')$ is the RPA spin susceptibility $\chi(\mathbf{k} - \mathbf{k}')$. For the models considered here, we find that the eigenvector $g_\alpha(\mathbf{k})$ corresponding to the largest eigenvalue $\lambda_\alpha$ has $s^\pm$ symmetry as shown in Fig. 6.

## Data availability

All data needed to evaluate the conclusions presented in this study have been deposited in Figshare database under the following accession code https://doi.org/10.6084/m9.figshare.25245994. The data for our TB calculations are available in the main text or the supplementary materials. Any additional data that support the findings of this study are available from the corresponding author upon request.

## Code availability

The Ab initio calculations are done with the code VASP. Simulation RPA codes are available from the corresponding author upon reasonable request.

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

## Acknowledgements

The work was supported by the U.S. Department of Energy (DOE), Office of Science, Basic Energy Sciences (BES), Materials Sciences and Engineering Division.

## Author contributions

Y.Z., L.-F.L., T.A.M. and E.D. designed the project. Y.Z., L.F.L. and T.A.M. carried out numerical calculations for DFT, the TB model, and RPA calculations. Y.Z., A.M., T.A.M. and E.D. wrote the manuscript. All co-authors provided useful comments and discussion on the paper.

## Competing interests

The authors declare no competing interests.
