## [Peer Review File · Nature Communications]

REVIEWER COMMENTS

Reviewer #1 (Remarks to the Author):

The manuscript “Structural phase transition, s+- wave pairing and magnetic stripe order in the bilayered nickelate superconductor La₃Ni₂O₇ under pressure” by Zhang et al. presents a theoretical combined DFT and tight-binding model Hamiltonian/RPA study of the superconducting properties and magnetic instabilities of bulk La₃Ni₂O₇. The impact of octahedral tilts is discussed on the basis of phonon calculations. Magnetic stripe order is suggested to play a role in this system.

The bilayer nickelates with their high T_c of about 80K under pressure attract considerable attention at the moment, so the paper is unambiguously timely and topical. I also find it very interesting, specifically the qualitative comparison of the new system to cuprates and iron-based superconductors. It will be a valuable addition to the literature.

On the other hand, the key results (specifically the s+- pairing) have been presented already earlier by various other groups, using an identical or comparable methodology (arXiv:2306.07275, arXiv:2306.03706, arXiv:2307.10144, arXiv:2307.14965 [cond-mat.supr-con]). Moreover, the employed methodology is straightforward and no new developments are presented here. Lastly, the study is purely theoretical. On the basis of all these aspects, it is legitimate to question whether Nature Communications with its very broad readership is the appropriate journal for this work, and I suggest rather publication in a more specialized journal.

In any case, I have some comments that need to be addressed by the authors before further consideration of the manuscript:

- Additional comparison with the infinite-layer nickelates would be interesting. In particular, I am wondering whether the techniques used by the authors can explain the pronounced difference in T_c between the infinite-layer nickelates and the pressurized bilayer Ruddlesden-Popper compound.
- In this context, additional data for an infinite-layer nickelate and a representative cuprate should be added to Fig. 7 to establish these aspects further, which is so far only qualitative.
- The authors base their study on DFT simulations with U = 4 eV. I consider this as an appropriate choice, but note that the electronic structure depends sensitively on this parameter in the present system, particularly the appearance of the ‘gamma’ hole pocket which is key in the s+- superconductivity mechanism. The authors should therefore investigate and discuss the impact of U. For instance, linear response suggests 6 eV [Sun et al.].

- While most simulations presented in the manuscript employ an effective $U = U - J$, the magnetic properties are assessed with separate U and J parameters. The authors should clarify which DFT approach is used in this case, and why. How does the band structure look like for the suggested stripe-ordered phase?
- The authors should visualize the scattering/pairing mechanisms for the magnetic susceptibility/superconductivity (add arrows, e.g., in Figs 3 and 6). Furthermore, the pressure values should be written in Fig. 6 for enhanced clarity.
- In line 264, the authors claim that their results imply nematicity. This is a rather crude statement and needs additional confirmation, or it should be made clear that it is purely speculative.
- Optimized/employed lattice parameters and a comparison to previous literature need to be provided.
- The authors are rather vague in describing their RPA methodology, in particular the U , U' , J , J' parameters that enter the formalism, which are distinct from the DFT values (I suppose). For the non-expert reader, these aspects remain unclear, and referring them to the extensive literature is only of limited value. It is also unclear where these parameters stem from and how sensitively the results depend on them. The Methods section needs to be a bit more detailed in this regard.
- Gpa should be replaced everywhere by GPa.

Reviewer #2 (Remarks to the Author):

This paper reports the results of theoretical calculations regarding superconductivity in $\text{La}_3\text{Ni}_2\text{O}_7$. While the physical properties of $\text{La}_3\text{Ni}_2\text{O}_7$ have been widely studied since the 1990s, recently, it was reported that $\text{La}_3\text{Ni}_2\text{O}_7$ becomes superconducting when pressure higher than $\sim 15\text{GPa}$ is applied after synthesizing pure single crystals. The superconducting transition temperature is astonishingly high ($\sim 80\text{K}$) and has attracted attention.

In general, to experimentally establish that a given material is a superconductor, the following four conditions must be met: observation of zero resistance, observation of the Meissner effect, identification of the crystal structure, and reproduction by independent groups. For $\text{La}_3\text{Ni}_2\text{O}_7$, signs of zero resistance were reported in Ref. 21, and zero resistance has been reported in arXiv:2307.14819 and arXiv:2307.09865. However, there has been no report of the Meissner effect. As pointed out by the reviewers of Ref. 21, the issue of inhomogeneity has not been resolved, and thus the identification of the crystal structure has not yet been completely solved. Moreover, all

samples were synthesized by M. Wang's group, and it cannot be said that reproduction by independent groups has been successful.

In this situation, it is premature to discuss the assumption that superconductivity is realized in La₃Ni₂O₇. Although Ref.21 was published in the highly prestigious journal, "Nature", it is clear from recent issues with hydride superconductors that being published in Nature does not prove that it is a superconductor. In this paper, more restrained and neutral expressions are required, at least in the introduction part. For example, it should be mentioned that just by measuring the resistance with a four-terminal measurement on a sample with unknown degrees of inhomogeneity, it is not possible to conclude whether it is superconducting.

With these points in mind, I will discuss the contents and results of the calculations in this paper.

(1) La₃Ni₂O₇ is expected to have strong instability of charge order. Although this study focuses on spin fluctuations, an equivalent discussion should also be made for the degree of freedom of charge. If the terminals for measuring voltage are insulating and the terminals for measuring current are metallic, it may appear to be superconducting. Verifying under what circumstances the instability of charge order in this system strongly appears is very important.

(2) A transition temperature of 80K is very high, even compared to the transition temperatures of copper oxide high-temperature superconductors. There should also be a discussion about whether, when the calculations made in this study are applied to LSCO, which has a lower transition temperature than 80K, the transition temperature of La₃Ni₂O₇ will be much higher than LSCO.

In conclusion, I do not think that the present version of the manuscript is suitable for publication in Nature Communications.

Reviewer #3 (Remarks to the Author):

The authors systematically study the electronic structure and magnetic properties of the newly discovered bilayer nickelate superconductor La₃Ni₂O₇ (LNO) as a function of pressure using first-principles (DFT+U) and RPA calculations. Using a phonon analysis, they show the Amam phase is stable at low pressures, while the Fmmm phase becomes stable approximately above 10 GPa where the experimental structural transition occurs. DFT+U calculations show that a stripe phase is the magnetic ground state of LNO at intermediate U and J and this conclusion is further supported by a

calculation of the RPA susceptibility derived from a four-orbital tight-binding model. Furthermore, the authors predict the dominant pairing of the gap for LNO to have s \pm symmetry.

Overall, the authors present a thorough study of LNO from a first-principles and tight-binding model perspective. However, there are a few issues that need to be addressed before considering the manuscript for publication in Nature Communications. These questions and comments are enumerated below.

1) The four orbital tight-binding (TB) model is derived by downfolding the Kohn-Sham bands onto Wannier orbitals and extracting the dominant hopping terms from this Wannier Hamiltonian. Do the TB dispersions actually reproduce the DFT bands? Compared to other theory works, there does not seem to be a van Hove singularity at the X point at zero pressure in the Fmmm phase. If this point is removed, then the RPA susceptibility suggesting stripe order will become greatly reduced if not disappear altogether.

2) Using DFT+U, the authors show the stripe (π , 0) phase to be the ground state at almost (or all) pressures. The authors choose the Dudarev double-counting scheme in their DFT+U calculations. In this scheme, the DFT+U energy functional has a single U_{eff} term ($U_{\text{eff}} = U - J$). The authors vary the Hund's coupling J , but keep U fixed to 4 eV. They show that for $U_{\text{eff}}=4-1=3\text{eV}$, $U_{\text{eff}}=4-0.8=3.2\text{eV}$ and $U_{\text{eff}}=4-0.6=3.4\text{ eV}$ the stripe phase is more stable. As shown in the supplement, for $U_{\text{eff}}=4-0.4=3.6\text{ eV}$, the G-type AFM phase is more stable instead. The authors conclude then that the stabilization of different magnetic states (at different U_{eff}) is driven by J . However, for the Dudarev double-counting scheme, the effect of U and J cannot be decoupled because J is not the coefficient in front of the anisotropic term, in other words, in Dudarev's approach the parameters U and J do not enter separately, only the difference $U-J$ is meaningful. How do the authors then interpret their results as being driven by J ? Can the authors show if their results (and particularly their main conclusions summarized above) are sensitive to the choice of U and/or to the choice of double counting scheme? For example, what happens within FLL where the effect of the anisotropy of the interaction can indeed be 'decoupled'?

3) For Figure 4 (where the RPA susceptibility is shown), what are the intraorbital and interorbital contributions to the susceptibility? At 0 GPa, what is the dominant contribution to the peak?

4) Minor point: in the introduction, the authors write that the bilayer nickelate LNO is the first non-infinite-layer nickelate to be superconducting, but this is not an accurate statement. After the infinite-layer nickelates (RNiO₂), the quintuple-layer nickelate (R₆Ni₅O₁₂) was found to be superconducting (Nature Materials 21, 160 (2021)). Perhaps the authors were referring to nickelates with infinite NiO₂ planes but then that sentence should be reworded for clarity.

Reviewer #4 (Remarks to the Author):

Response to the first Referee

We thank the first referee for his/her careful reading, valuable comments, suggestions, and recommendations for our work. This referee writes “*The bilayer nickelates with their high T_c of about 80K under pressure attract considerable attention at the moment, so the paper is unambiguously timely and topical. I also find it very interesting, specifically the qualitative comparison of the new system to cuprates and iron-based superconductors. It will be a valuable addition to the literature.*” Below, and in the revised text, please find our answer to the issues raised by this referee.

Comment: On the other hand, the key results (specifically the s_{\pm} pairing) have been presented already earlier by various other groups, using an identical or comparable methodology (arXiv:2306.07275, arXiv:2306.03706, arXiv:2307.10144, arXiv:2307.14965 [cond-mat.supr-con]). Moreover, the employed methodology is straightforward, and no new developments are presented here. Lastly, the study is purely theoretical. On the basis of all these aspects, it is legitimate to question whether Nature Communications with its very broad readership is the appropriate journal for this work, and I suggest rather publication in a more specialized journal.

Response: We disagree with this comment by this referee. First, the four competing publications mentioned by this referee, and our present paper (arXiv 2307.15276), as well as others after ours, were done individually after the original experimental paper was initially posted in arXiv. It is natural that after a very important experimental result is posted, that theorists start eagerly working on the subject.

The “first” study that appears in the arXiv does not mean priority to claim novelty. In other words, it does not mean that other competing studies should be diminished. Furthermore, this is also supported by the policy of novelty in Nature Communications. On the contrary, those parallel papers, before and after ours, do support the reliability of the s_{\pm} pairing conclusions by using different many-body methods. Their existence also shows with clarity the “general interest” character of the subject.

Furthermore, the first three papers have already been cited in our present manuscript, but the fourth paper (arXiv 2307.14965) only appeared on arXiv after our initial submission to Nature Communications, so we did not notice this paper at the moment. In addition, the four preprints mentioned by this referee focused on discussing s_{\pm} wave only for one specific pressure. *However, in our study, we considered a broad pressure range.* Thus, our study obviously needed far more computing resources and time, leading to a slightly late

submission, compared to the first paper mentioned by the referee. It cannot be that whoever publishes first in the arXiv wins: many factors of the work should also matter.

Moreover, we were the first to theoretically demonstrate the structural instability and the first-order nature of the phase transition at the mysterious range of 10 to 14 GPa. We also discussed the magnetic stripe order under pressure, which was not discussed in the papers mentioned by the referee. Finally, theoretical studies are part of the scope of Nature Communications, where there are plenty of purely theoretical papers published. It is not mandatory to have an experimental component.

In summary, this comment from the first referee is not reasonable. Our study provides explanations for the pressure-dependent properties of LNO, including the mysterious range 10-14 GPa, and formulates predictions for incipient magnetic order and pairing channels, as well as the importance of the γ pocket. We also show that LNO is qualitatively different from BSCCO, thus it deserves publication in Nature Communications even if other, less complete, preprints appeared first by just a few weeks in the arXiv.

To highlight the importance and novelty of our work to readers, we also added and modified several sentences in the revised version.

Comment 1: Additional comparison with the infinite-layer nickelates would be interesting. In particular, I am wondering whether the techniques used by the authors can explain the pronounced difference in T_c between the infinite-layer nickelates and the pressurized bilayer Ruddlesden-Popper compound.

Response: This is a good question raised by this referee. In fact, many studies have provided different perspectives and ideas to understand superconductivity in those infinite-layer nickelates for the bulk format. For example, *d*-wave superconductivity, analogous to the cuprate superconductors, was proposed in this system (see X. Wu et al. PRB101-060504(R)-2020, H. Sakakibara et al. PRL125-077003-2020, and many others). In addition, the minimum model of the infinite-layer nickelate systems for the superconductivity is still under debate, such as the one-single x^2-y^2 orbital model [see PRB101-060504(R)-2020 as an example], two e_g (x^2-y^2 and $3z^2-r^2$) orbitals model [See PRL124-207004-2020], two-band model (x^2-y^2 and xy) [PRR1-032046R-2019], and many others. Furthermore, to our best knowledge, superconductivity was not observed in bulk [see Q. Li et al. Commun. Mater. 1 16 (2020) as an example] in the infinite-layer nickelate systems.

In our previous publication (PRB102-195117-2020), we also discussed the possible interface

effects of two two-dimensional electron gas in the superconductivity of infinite-layer nickelates. Moreover, hydrogen was also considered to have a critical role in the superconductivity of infinite-layer nickelates in some studies [see L. Si et al. PRL124-166402-2020, X. Ding et al. Nature 615 50 (2023) and many others.]. For those reasons, we do not have yet a good answer to this comment. It still needs more careful studies in detail, namely, not all the many issues of this family of compounds can be addressed in one paper. At this point a historical remark is worth formulating: in the Cu-based high- T_c superconductors it is well known that the T_c of single layers is much less than bilayers, which are themselves less than trilayers. *However, after decades of work, there is still no clear answer to why this happens.* Probably a lot of time will pass until the question of the referee is answered in nickelates.

From our perspective, tentatively we believe that the main difference between bilayers and infinite layers is that the $3z^2-r^2$ orbital plays quite different roles in the infinite-layer nickelates and bilayer $\text{La}_3\text{Ni}_2\text{O}_7$. Specifically, the $3z^2-r^2$ orbital forms a bonding-antibonding state, leading to a larger intraorbital hopping between the two Ni layers, and eventually leading to the s_{\pm} -wave superconductivity. However, *this bonding-antibonding state was not obtained in the infinite-layer nickelates*, resulting in a weaker inter-layer coupling. In addition, a sizable interorbital hopping between e_g orbitals was found in the bilayer $\text{La}_3\text{Ni}_2\text{O}_7$, while this value is nearly zero in the infinite-layer nickelates, leading to in-plane stripe vs G-AFM in those two systems, respectively. These two characteristics are the main differences between the two systems.

Accordingly, in the revised version we added some sentences addressing the discussion of infinite-layer nickelates from our perspective, to highlight the main differences between those two systems. We thank this referee for the interesting comment about infinite-layer nickelates, which helped us highlight the uniqueness of the bilayer LNO system.

Comment 2: In this context, additional data for an infinite-layer nickelate and a representative cuprate should be added to Fig. 7 to establish these aspects further, which is so far only qualitative.

Response: This is an interesting comment by this referee. We guess that this referee probably wants us to discuss the differences between LNO and the infinite-layer nickelates, as well as Cu-based superconductors to explain the quantitative differences in T_c . This comment raised by this referee is very interesting, but it is unrealistic for a single paper.

For the observed superconductivity in the infinite-layer nickelates, there are several different perspectives and ideas. Even the minimum model is still under hot debate. At present, both theoretical and experimental works continue trying to understand the superconductivity in infinite-layer nickelates and many new results are published from time to time. People including us still do not know the “truth” of superconductivity in the infinite-layer nickelates. However, science is always done this way: we refine and refine, following the initial discovery.

In the cuprates, superconductivity is believed to be driven by the in-plane antiferromagnetic (AFM) fluctuations of the x^2-y^2 orbital, resulting in a d -wave superconductivity. In addition, the inter-layer coupling is weak in the cuprates, leading to an “effective” single-layer system. However, instead, we found s_{\pm} -wave superconductivity driven by inter-layer (AFM) fluctuations in LNO. Hence, the main difference is the importance of the $3z^2-r^2$ orbital, leading to one key orbital vs two key orbitals in the cuprates and bilayer LNO superconductors, respectively. Furthermore, oxygen also plays a different role in the bilayer LNO and cuprates superconductors. LNO has a larger charge-transfer gap from oxygen p to Ni’s $3d$ orbitals than cuprates. Thus, LNO is a Mott-Hubbard system while cuprates are a charge-transfer system. This is also a fundamental difference between nickelate and cuprates superconductors.

To address the many comments of this referee, we have tried hard the challenge of comparing LNO vs. the typical bilayer BSCCO. But this is too much by using only a single orbital x^2-y^2 model. In order to get the same λ for the single-band model for BSCCO at a filling of $n=0.85$, we need a 25% larger U , which seems to be reasonable because cuprates have stronger electronic correlations than nickelates. However, note that a similar λ doesn’t necessarily translate to the same T_c . Since $T_c \sim w_0 \exp(-1/\lambda)$, the cut-off frequency prefactor w_0 also plays a role, which is obviously different in these two systems.

In addition, to provide a good comparison in T_c between LNO and a typical cuprate, many complicated effects and more orbitals also need to be considered in the model calculations, which requires far more efforts that cannot be done in a short time. Thus, we found that there are too many variables to make a conclusive statement for this demanding comment. Note that the current works of LNO are on the qualitative level, not on the quantitative level, not to mention including quantitative comparisons with cuprates. Not all the many issues can be fully addressed in one paper.

For these reasons, we can only provide qualitative ideas for comparing LNO vs. infinite-layer nickelates, as well as cuprate superconductors. Accordingly, in the revised version we added

some paragraphs about the interesting discussion raised by this referee. We also thank this referee for the interesting comment about infinite-layer nickelates and cuprates, which helped us highlight the uniqueness of the bilayer LNO system.

Comment 3: The authors base their study on DFT simulations with $U = 4$ eV. I consider this as an appropriate choice but note that the electronic structure depends sensitively on this parameter in the present system, particularly the appearance of the ‘gamma’ hole pocket which is key in the $s\pm$ superconductivity mechanism. The authors should therefore investigate and discuss the impact of U . For instance, linear response suggests 6 eV [Sun et al.].

Response: This is an interesting comment by this referee. However, we do not see any problem with our choice of $U = 4$ eV. The recent *ab initio* study estimates U is about 3.8 eV obtained from the constrained random phase approximation [see V. Christiansson et al. arXiv 2306.07931 (2023)]. In addition, our results found the Fmmm phase becomes stable around 10.6 GPa whereas the experimental structural transition occurs around 10 GPa [H. Sun et al. Nature 621 493 (2023), Y. Zhang et al. arXiv 2307.14819 (2023)], denoting good agreement.

Furthermore, in the paper mentioned by this referee [H. Sun et al. Nature 621 493 (2023)], they also discussed the DFT results in the main text by using $U = 4$ eV. Moreover, following the request by this referee, our optimized lattice parameters of the Amam phase without pressure for $U = 4$ eV and $U = 6$ eV are $a/b/c = 5.433/5.367/20.670$ Å and $a/b/c = 5.366/5.299/21.429$ Å, respectively. For comparison, previous neutron experiments reported that the crystal lattices are $a = 5.448$ Å, $b = 5.393$ Å, and $c = 20.518$ Å, while recent X-ray diffraction found the lattice constants are $a = 5.438$ Å, $b = 5.400$ Å, and $c = 20.455$ Å. Of course, the agreement in the calculation of structural parameters does not provide a solid foundation to support the choice of U .

However, from the accumulation of all the results expressed here the choice of $U = 4$ eV appears better than $U = 6$ eV. Considering those facts, we do not think our choice of $U = 4$ eV has any serious weaknesses.

To better help readers understand this point, we revised and added sentences to address the choice of $U = 4$ eV in the “DFT method” section.

Comment 4: While most simulations presented in the manuscript employ an effective $U = U - J$, the magnetic properties are assessed with separate U and J parameters. The authors should clarify which DFT approach is used in this case, and why. How does the band structure look like for the suggested stripe-ordered phase?

Response: This is a good comment by this referee. Typically, intraorbital hopping leads to antiferromagnetic (AFM) coupling between two half-filled orbitals, resulting in the canonical AFM Heisenberg superexchange interaction. However, the importance of the interorbital hopping favoring ferromagnetic (FM) coupling between half-filled and empty orbitals via Hund's coupling J , was recently discussed in our previous studies [see L.-F. Lin et al. PRL127-077204-2021, Commun. Phys. 6 199 (2023)]. In the case of LNO, we believe the in-plane magnetic stripe order, with a mixture of FM bonds in one direction and AFM bonds in the other direction, can be understood by the strong competition between intraorbital and interorbital hopping mechanisms, namely the competition of AFM and FM tendencies explained above.

Here, we would like to discuss the effect of J on the magnetic stripe instability. However, the U and J are not decoupled in the Dudarev scheme, where only a single U_{eff} term ($U_{\text{eff}} = U - J$) works. Hence, to understand the importance of Hund coupling J , we used the Liechtenstein formulation within the double-counting item, where U and J are decoupled (previously cited as Ref. [53] and now Ref. [56] in our discussion of the magnetic states). All this information is now in the revised version.

To avoid confusing the readers, we revised sentences and added extra sentences to clarify the issue of U and J in both the "stripe order instability" and the "DFT method" sections of the revised version. In addition, following the request of this referee, we also now show the band structures of the magnetic stripe phase from $J = 1$ eV to $J = 0.4$ eV at 25 GPa in the supplemental materials.

Comment 5: The authors should visualize the scattering/pairing mechanisms for the magnetic susceptibility/superconductivity (add arrows, e.g., in Figs 3 and 6). Furthermore, the pressure values should be written in Fig. 6 for enhanced clarity.

Response: Thanks for these good suggestions by this referee. Accordingly, we added the sketch of the magnetic configurations as a panel inset in Fig. 4 and the signs of the superconducting parameters in Fig. 6, as well as added the pressure values in Fig. 6.

Comment 6: In line 264, the authors claim that their results imply nematicity. This is a rather crude statement and needs additional confirmation, or it should be made clear that it is purely speculative.

Response: This is a good comment by the referee. In the context of the study of iron-based superconductors, the “nematicity” was related to the in-plane stripe $(\pi, 0)$ spin order [see PRB81-140501(R)-2010, PRL111-047004-2013 as examples]. By mere analogy, note that if the “nematicity” of Fe-based pnictides and selenides superconductors is driven by stripe magnetic order as widely assumed, it is very natural to think the same may occur in the case of LNO as well. Of course, this topic deserves further investigation and detailed discussion. Accordingly, we also added and revised some sentences in the revised version to help readers understand the context and to avoid misleading them. Several references of nematicity in iron superconductors were added.

Basically, the idea of nematicity is that in principle both the $k_1 = (\pi, 0)$ and $k_2 = (0, \pi)$ stripe orders must be *degenerate*. I.e., it was discussed long ago that for this reason, it may occur that upon lowering the temperature, first a $Z(2)$ Ising symmetry breaking occurs at $T = T^*$ where the spin structure factor of k_1 and k_2 are no longer equal for lower temperatures. But this breaking is nematic, namely does not involve long-range order. Then, upon lowering even further the temperature, the $O(3)$ symmetry is finally broken, and long-range order develops at a second transition. The region in between the two critical temperatures is the nematic regime and was widely discussed in iron superconductors. In the revised version, we added several sentences about this topic.

Comment 7: Optimized/employed lattice parameters and a comparison to previous literature need to be provided.

Response: This is an interesting comment. To our best knowledge, all the present theoretical studies of this bilayer nickelate use the experimental crystal constants or simply do not provide detailed information of lattice parameters.

Thus, following the comment by this referee, we just compared our optimized lattice parameters vs. the structural parameters obtained from different experimental groups for the Amam phase at ambient conditions. More specifically, our optimized lattice parameters are $a = 5.434 \text{ \AA}$, $b = 5.367 \text{ \AA}$, and $c = 20.670 \text{ \AA}$, which are in good agreement with the experimental values such as in a previous neutron scattering study ($a = 5.448 \text{ \AA}$, $b = 5.393 \text{ \AA}$, and $c = 20.518 \text{ \AA}$) at 300 K and recent X-ray diffraction at 300 K ($a = 5.438 \text{ \AA}$, $b = 5.400 \text{ \AA}$,

and $c = 20.455 \text{ \AA}$). Accordingly, we added those results in the “DFT method” section of the revised manuscript.

Comment 8: The authors are rather vague in describing their RPA methodology, in particular the U, U', J, J' parameters that enter the formalism, which are distinct from the DFT values (I suppose). For the non-expert reader, these aspects remain unclear, and referring them to the extensive literature is only of limited value. It is also unclear where these parameters stem from and how sensitively the results depend on them. The Methods section needs to be a bit more detailed in this regard.

Response: Thanks for this interesting comment. To help readers better understand our calculations, we now introduced in the text the full Hamiltonian, including the Kanamori Hubbard interactions, for the bilayer Hubbard model in the portion of RPA of the “Method” section. Furthermore, we also define the operators and each term explicitly. Moreover, we defined the relations between couplings due to rotational invariance. When these relations are used, only two parameters remain: the Hubbard U and the Hund coupling.

Comment 9: Gpa should be replaced everywhere by GPa.

Response: Thanks for bringing up this issue to our attention. We carefully addressed those typos in the revised manuscript, plus we used the opportunity to reread and revise several times the presentation of the entire manuscript.

In our opinion, we have properly answered all the comments and requests of the first referee and our paper should now be ready for publication.

Response to the Second Referee

We thank the second referee for his/her valuable comments, suggestions, and recommendations for the publication of our work.

General comment: In general, to experimentally establish that a given material is a superconductor, the following four conditions must be met: observation of zero resistance,

observation of the Meissner effect, identification of the crystal structure, and reproduction by independent groups. For $\text{La}_3\text{Ni}_2\text{O}_7$, signs of zero resistance were reported in Ref. 21, and zero resistance has been reported in arXiv:2307.14819 and arXiv:2307.09865. However, there has been no report of the Meissner effect. As pointed out by the reviewers of Ref. 21, the issue of inhomogeneity has not been resolved, and thus the identification of the crystal structure has not yet been completely solved. Moreover, all samples were synthesized by M. Wang's group, and it cannot be said that reproduction by independent groups has been successful.

In this situation, it is premature to discuss the assumption that superconductivity is realized in $\text{La}_3\text{Ni}_2\text{O}_7$. Although Ref.21 was published in the highly prestigious journal, "Nature", it is clear from recent issues with hydride superconductors that being published in Nature does not prove that it is a superconductor. In this paper, more restrained and neutral expressions are required, at least in the introduction part. For example, it should be mentioned that just by measuring the resistance with a four-terminal measurement on a sample with unknown degrees of inhomogeneity, it is not possible to conclude whether it is superconducting.

Response: The comments of the referee are correct. Experimentally, the best complete evidence of superconductivity should arise from both resistivity measurements and the Meissner effect. As this referee said, while zero resistance has been reported by experimental works, the Meissner effect has not been observed yet.

M. Wang's group observed a sharp transition and flat stage in resistance by using KBr as the pressure-transmitting medium, as well as a diamagnetic response in susceptibility, which they believe originated from the two prominent properties of superconductivity, zero resistance, and Meissner effect. We are in direct contact with them, and we confirmed these statements.

Recently, the potentially "filamentary" nature of the superconducting state has been presented by an experimental group (paper not yet in arXiv; we received a draft for our comments, thus we cannot provide the full reference). This filamentary perspective provides a tentative explanation for why the Meissner effect has not been observed yet. This is basically a matter of sample quality, namely the inhomogeneities the referee correctly mentions. In our opinion, as the sample preparation improves, the Meissner effect will eventually be unveiled. Also, it is important to remark that the paper mentioned above (not yet in arXiv) and a new paper from Y. Qi's group (arXiv:2309.01651), both correspond to groups different from M. Wang's group. Namely, reproduction by independent groups has been successful.

For the reasons explained above, the reproduction of results by other groups confirms the original claims of superconductivity in LNO at high pressure, at least with regard to resistivity.

We totally agree that the issues with hydride superconductors are of potential concern, but by now the results of M. Wang's group have been reproduced by others so the situation in nickelates is more robust than that in hydride superconductors.

However, we have no problem at all in adding more restraint and neutral expressions as the referee proposes. This is wise to do. Then, in the revised version we also added and revised sentences in the "Introduction" section to alert the readers that the Meissner effect still has not been found and that inhomogeneities in the sample are a potential reason for concern.

Comment 1: $\text{La}_3\text{Ni}_2\text{O}_7$ is expected to have strong instability of charge order. Although this study focuses on spin fluctuations, an equivalent discussion should also be made for the degree of freedom of charge. If the terminals for measuring voltage are insulating and the terminals for measuring current are metallic, it may appear to be superconducting. Verifying under what circumstances the instability of charge order in this system strongly appears is very important.

Response: This is a very good comment by this referee. In the experimental paper [H. Sun et al. Nature 621 493 (2023)], M. Wang's group has excluded the possibility of the presence of charge-density-wave order at low temperatures under pressures from 14.0 ~ 43.5 GPa, based on the resistance measurements. Furthermore, other experimental studies related to this compound $\text{La}_3\text{Ni}_2\text{O}_7$ also did not provide any evidence of a charge density wave yet.

Motivated by the interesting comments of the referee, we have carried out DFT studies of charge order instabilities in $\text{La}_3\text{Ni}_2\text{O}_7$. We considered a checkerboard charge order (G-type) configuration with both antiferromagnetic coupling in-plane and between two Ni layers to facilitate our work. This checkerboard charge order was also proposed in another bilayer nickelate $\text{La}_3\text{Ni}_2\text{O}_6$ [A. S. Botana et al. PRB94 081105(R) (2016)]. Here, we studied two specific pressure values (15 GPa and 30 GPa) in the Fmmm phase, where the two values are in the "superconducting" phase diagram in the experiment [H. Sun et al. Nature 621 493 (2023)]. Furthermore, we used the specific values $U = 4$ eV and $J = 0.6$ eV, very similar to those obtained from constrained density functional calculations ($U \sim 3.8$ eV and $J \sim 0.61$ eV) [see V. Christiansson et al. arXiv 2306.07931 (2023)].

Specifically, we observed a robust charge disproportionation in this charge order state with 0.933 and 0.377 μ_B/Ni for two different Ni sites, but this state has a higher energy (about ~ 8.11 meV/Ni) than the magnetic stripe state at 15 GPa that we studied. Moreover, we do not obtain any obvious charge disproportionation at 30 GPa. In addition, we also considered

structural distortions, namely lattice relaxations, and our conclusions remain the same. Increasing J to 0.8 eV, by introducing the structural distortions, the charge disproportionation increases to 1.236/0.364 and 0.953/0.395 μ_B/Ni , for 15 and 30 GPa, respectively, in this charge order state. Although the charge disproportionation is enhanced in this charge order state with higher J , the enthalpy is still higher (~ 17.47 and 20.89 meV/Ni) than the enthalpy in the magnetic stripe state at 15 and 30 GPa. *Hence, from our DFT perspective, the charge density wave state is not stable in the $Fm\bar{3}m$ phase under pressure.*

Accordingly, we also added some sentences to briefly discuss this very interesting issue in the “Charge order instability” among the “Results” of the revised version. We thank the referee for the interesting insights into the instability of the charge order which highly improves the quality of our publication.

Comment 2: A transition temperature of 80 K is very high, even compared to the transition temperatures of copper oxide high-temperature superconductors. There should also be a discussion about whether, when the calculations made in this study are applied to LSCO, which has a lower transition temperature than 80K, the transition temperature of $\text{La}_3\text{Ni}_2\text{O}_7$ will be much higher than LSCO.

Response: Probably the referee would like us to discuss the connections between LNO and previously well-studied Cu-based superconductors. In the cuprates, superconductivity is believed to be driven by the in-plane antiferromagnetic (AFM) fluctuations of the x^2-y^2 orbital, resulting in a d -wave superconductivity. In addition, the inter-layer coupling is weak in the cuprates, leading to an “effective” single-layer system.

However, instead, in LNO we found $s\pm$ -wave superconductivity driven by inter-layer (AFM) fluctuations. Furthermore, in the context of the study of cuprates, T_c increases substantially going from 1 layer to 2 layers, and then slightly more for 3 layers [see PRM2-123401-2018 as an example]. But the Ruddlesden-Popper (RP) Ni-oxide layered materials do not simply following the same trend as the cuprates: superconductivity was not observed in single-layer and trilayer RP nickelates at the pressure region they studied [see arXiv 2309.01651 (2023)]. Interestingly, the inter-layer coupling is also much weaker in the single-layer and tri-layer RP nickelates, suggesting the importance of inter-layer coupling for the superconductivity in Ni-oxide bi-layered materials. We are working on this topic to provide an explanation for the absence of superconductivity in the single-layer and tri-layer RP nickelates. Work is in progress and will be reported soon.

In addition, oxygen also plays a different role in the bilayer LNO and cuprates superconductors. LNO has a large charge-transfer gap from oxygen p to Ni's $3d$ orbitals, like the infinite-layer nickelate superconductors, indicating a Mott-Hubbard system. However, this charge-transfer gap is quite smaller in cuprates superconductors, leading to a charge-transfer system. This is also a fundamental difference between nickelate and cuprates superconductors.

In the context of the study of cuprates, the x^2-y^2 orbital is widely believed to play a key role in superconductivity. Here, we have tried hard but the challenge of comparing LNO and the typical bilayer BSCCO (has a T_c of 84 K without the need of adding pressure) is too much by using a single-orbital x^2-y^2 model. In order to get the same λ for the single-band model for BSCCO at a filling of $n = 0.85$, we need a $\sim 25\%$ larger U , which seems to be reasonable because cuprate has stronger electronic correlations than nickelate. Note that a similar lambda does not necessarily translate to the same T_c . Because $T_c \sim w_0 \exp(-1/\lambda)$, the cut-off frequency prefactor w_0 also plays a role, which is different in those two systems. To provide a good comparison in T_c between LNO and a typical cuprate, many complicated effects and more orbitals need to be considered in the model calculations. All in all, we found that there are too many variables to make a conclusive statement, which needs unrealistic efforts beyond the scope of this publication.

Hence, in the present study, we can only provide some qualitative ideas for comparing the differences between the LNO and cuprate superconductors. Accordingly, we added some paragraphs for this interesting discussion raised by this referee in the revised version, which is enough for the readers to understand the difference between those two systems.

In our opinion, we have properly answered all the comments and requests of the second referees and our paper should now be ready for publication.

Response to the Third Referee

We thank the third referee for his/her careful reading and valuable comments. This referee writes “*Overall, the authors present a thorough study of LNO from a first principles and tight-binding model perspective. However, there are a few issues that need to be addressed before considering the manuscript for publication in Nature Communications...*” Below, and

in the revised text, we have addressed all the questions/comments of this referee.

Comment 1: The four-orbital tight-binding (TB) model is derived by downfolding the Kohn-Sham bands onto Wannier orbitals and extracting the dominant hopping terms from this Wannier Hamiltonian. Do the TB dispersions actually reproduce the DFT bands? Compared to other theory works, there does not seem to be a van Hove singularity at the X point at zero pressure in the Fmmm phase. If this point is removed, then the RPA susceptibility suggesting stripe order will become greatly reduced if not disappear altogether.

Response: This is also a good comment by the referee. Yes, we agree with this referee that the stripe order would be considerably reduced if the vHs moves away from the Fermi level. In fact, this is also supported by our RPA and DFT+ $U+J$ results. As pressure increases, the vHs shifts down from the Fermi level in the TB band structures. Then, based on our RPA calculations, the peak at $(\pi, 0)$ of RPA susceptibility strongly reduces as the pressure increases, as shown in Fig. 5. Furthermore, in our DFT+ $U+J$ calculations, the energy difference between the stripe and G-AFM also strongly decreases as the pressure increases (see Fig. 4). This robust reduced energy difference indicates that the strong competition between FM and AFM in the plane direction enhances under pressure. As we discussed in the main text, we believe those results suggest that the long-range spin stripe order is not possible in the Fmmm phase under high pressure.

Having the vHs *exactly* at the Fermi level can only happen by chance. But having the vHs “close” to the Fermi level can be important. Whether this is a true singularity or just a broadened sharp peak cannot be important for superconductivity. A small shift of the flat band at X from the Fermi level cannot change the physics much. For the context of the previous well-addressed cuprates superconductor, there is also a similar vHS at the same X point but the community does not think this is a fundamental issue. Yes, here the vHs “happens” at the X point in our TB band with nearest-neighbor hopping, but this singularity would be softened by adding additional hoppings beyond nearest-neighbors, or by other means such as broadening by disorder.

In the high-pressure region, this vHs shifts down from the Fermi level. So, it is not crucial whether we are at 0 or 50 GPa to discuss the vHs in LNO. However, the value of λ we reach at ambient pressure is much enhanced by the vHs. Here, the important physics is to have the Fmmm phase for superconductivity and, more accurately, have the γ pocket. As we discussed in the main text, the superconductivity pairing tendency was strongly suppressed when the γ pocket disappeared even though the vHS is still “exactly” at the X point in our TB band structure. Furthermore, our present study also indicates that the Fmmm phase is not stable in

the low-pressure range, as well as the experimental result. In addition, our very recent efforts show that the substrate or strain could not stabilize the Fmmm phases (not published yet) at 0 GPa. However, to avoid confusing readers with too much detail, we removed some sentences about this issue in the revised text.

Accordingly, we revised and added some sentences to remark the issue of vHs to help readers better understand this point in the revised version, as well as the discussion of stripe order. Moreover, we lower the emphasis on the vHs and soften this point in the revised version as well. We really thank the referee for the interesting insights provided that improve the quality of our publication.

Comment 2: Using DFT+U, the authors show the stripe ($\pi, 0$) phase to be the ground state at almost (or all) pressures. The authors choose the Dudarev double-counting scheme in their DFT+U calculations. In this scheme, the DFT+U energy functional has a single U_{eff} term ($U_{\text{eff}} = U - J$). The authors vary the Hund's coupling J , but keep U fixed to 4 eV. They show that for $U_{\text{eff}} = 4 - 1 = 3\text{eV}$, $U_{\text{eff}} = 4 - 0.8 = 3.2\text{eV}$ and $U_{\text{eff}} = 4 - 0.6 = 3.4\text{ eV}$ the stripe phase is more stable. As shown in the supplement, for $U_{\text{eff}} = 4 - 0.4 = 3.6\text{ eV}$, the G-type AFM phase is more stable instead. The authors conclude then that the stabilization of different magnetic states (at different U_{eff}) is driven by J . However, for the Dudarev double-counting scheme, the effect of U and J cannot be decoupled because J is not the coefficient in front of the anisotropic term, in other words, in Dudarev's approach the parameters U and J do not enter separately, only the difference $U - J$ is meaningful. How do the authors then interpret their results as being driven by J ? Can the authors show if their results (and particularly their main conclusions summarized above) are sensitive to the choice of U and/or to the choice of double counting scheme? For example, what happens within FLL where the effect of the anisotropy of the interaction can indeed be 'decoupled'?

Response: This is also a good comment by this referee and thanks for bringing up this incomplete description of the DFT+U+J calculations to our attention. In the study of crystal structures under pressure, we used the Dudarev formation with $U_{\text{eff}} = U - J = 4\text{ eV}$ following the choice of the previous DFT study of $\text{La}_3\text{Ni}_2\text{O}_7$ [H. Sun et al. Nature 621 493 (2023)]. As the referee said, the U and J are not decoupled in the Dudarev scheme because only a single U_{eff} term ($U_{\text{eff}} = U - J$) is used. Hence, to understand the importance of the Hund coupling J , we used the DFT plus U and J with the Liechtenstein formulation within the double-counting item, previously cited as Ref. [53] and now as Ref. [56], in our discussion of the magnetic

states.

To avoid misleading the readers, we revised and added some sentences to clarify the issue of U and J in both the “Stripe order instability” and “DFT method” sections of the revised version.

Comment 3: For Figure 4 (where the RPA susceptibility is shown), what are the intraorbital and interorbital contributions to the susceptibility? At 0 GPa, what is the dominant contribution to the peak?

Response: This is also a good comment by the referee. For our RPA study, the susceptibility of Fig. 5 is renormalized, and all the contributions are mixed due to the matrix form of the RPA equations. To see which contributions are dominant, we look at χ^0 , where this is the bare unrenormalized susceptibility. Below are the results for the orbital contributions of χ^0 at 0 GPa.

As shown in the black curve, χ^0 has two peaks, one close to $(\pi, 0)$ and another one between $(\pi, 0)$ and (π, π) . Then, the $(\pi, 0)$ peak would strongly enhance while the other one between $(\pi, 0)$ and (π, π) would not, when taking into account electronic interactions in the RPA. In addition,

we also plotted the dominant contributions of the three orbitals (see the blue, red, and green curves). Adding them up gives a result very close to the black χ^0 curve. Here, the orbitals are indexed as “0” for the x^2-y^2 orbital in layer1, “1” for the $3z^2-r^2$ orbital in layer1, “2” for the x^2-y^2 orbital in layer2, and “3” for the $3z^2-r^2$ orbital in layer2. Clearly the scattering between the $3z^2-r^2$ orbitals dominates, and the inter-orbital scattering between the layers (1133) is almost as strong as the intra-orbital $3z^2-r^2$ scattering within a layer.

In addition, we also show the diagonal and off-diagonal contributions to the RPA spin susceptibility. Here, the off-diagonal contribution to the peak around $(\pi, 0)$ is just as large as the diagonal one due to the $3z^2-r^2$ scattering between the two layers.

Accordingly, we also added those results into the supplemental materials.

Comment 4: Minor point: in the introduction, the authors write that the bilayer nickelate LNO is the first non-infinite-layer nickelate to be superconducting, but this is not an accurate statement. After the infinite-layer nickelates ($RNiO_2$), the quintuple-layer nickelate ($R_6Ni_5O_{12}$) was found to be superconducting (Nature Materials 21, 160 (2021)). Perhaps the authors were referring to nickelates with infinite NiO_2 planes but then that sentence should be reworded for clarity.

Response: Thanks for pointing out this inaccurate description in our paper. Yes, we wanted to say that LNO is the first non-IL NiO₂-plane layered nickelate superconductor. Accordingly, we revised this description and added one sentence to discuss the quintuple-layer nickelate (R₆Ni₅O₁₂). In addition, we also used the opportunity to reread and revise several times the presentation of the entire manuscript.

In our opinion, we have properly answered all the comments and requests of the third referee and incorporated many useful sentences into the revised version, and our paper should now be ready for publication.

Response to the Fourth Referee

We thank the fourth referee for his/her careful reading and valuable comments. This referee writes *“I co-reviewed this manuscript with one of the reviewers who provided the listed reports. This is part of the Nature Communications initiative to facilitate training in peer review and to provide appropriate recognition for Early Career Researchers who co-review manuscripts.”*

=====

Summary of the main changes

1. All the main changes introduced in the main text have been marked in blue.
2. All the “Gpa” statements have been revised to “GPa” in the revised version. This is not marked in blue.
3. Seven references, Refs. (21-24, 40, 49, 59-60, 66), were added. In addition, some references cited in the previous version were removed to match the limitation of 70 references in Nature Communications and the order of references was renumbered.
4. The supplemental material (SM) was revised, and the main changes have been marked in blue.

REVIEWER COMMENTS

Reviewer #1 (Remarks to the Author):

The authors have responded convincingly to most of my comments, and made a great effort to enhance the manuscript. This will certainly be helpful to the reader. However, two aspects remain open:

- Comment 2 of Referee 1: I think that the new passages in the text are a valuable addition. The point of this comment, which also relates to Comment 2 of Referee 2, is to assess the reliability of the methodology across different materials. I can understand the arguments put forward by the referees. Nevertheless, additional results, maybe on λ instead of T_c to provide trends, would have been insightful – particularly for a journal with broader readership.

- Comment 3 of Referee 1: The authors rationalized their choice of U . I think that it is important to stress in the paper that variations of U may result in qualitatively different results. A good paper should investigate the robustness of the key results as a function of U . I think that this is particularly relevant here since, as Referee 2 highlighted, it is not clear whether these materials are really superconducting or not.

Reviewer #2 (Remarks to the Author):

The authors have appropriately responded to all the comments and criticisms of the reviewers. I think that the paper is now suitable for publication in Nature Communications.

Reviewer #3 (Remarks to the Author):

We would like to thank the authors for their work in answering our questions and improving the manuscript.

We are just left with a couple of points on which we would like further clarification on their end. Now that the authors have stated that the energy curves are obtained within FLL, we would like for them to clarify why the results here are different from those shown in their previous work. In Phys. Rev. B 108, L180510 (2023) the order (from lower to higher energy) at the same U and J values shown here is: stripe, A-type AFM, G-type AFM. Here, at the same value of U and J the order changes to stripe, G-type AFM, A-type AFM. Granted that the stripe is still the ground state but for reproducibility, the authors should clarify where the difference between the two sets of calculations is coming from.

Concerning the presence of the vHS at the Fermi level, toning down the statements related to 1) the nesting and 2) the pairing might not be sufficient. On 1) If one were to shift the Fermi energy far from the van Hove singularity (in the DFT calculations is ~ 0.2 eV above it) the susceptibility will for sure change and whether the peaks hinting a nesting for CDW formation remain is not clear (the authors do get a stripe as the ground state and maybe it is best to just say that if the vHS turns out to be crucial for the nesting features in the susceptibility). On 2) perhaps such a shift for the pairing will remove the 'artifact' of λ for pairing at ambient pressure and hence give rise to more realistic results.

Reviewer #4 (Remarks to the Author):

I co-reviewed this manuscript with another reviewer. My comments have been incorporated into this report.

Response to the first Referee

We thank the first referee for his/her careful reading, valuable comments, suggestions, and recommendations for our work. This referee writes “*The authors have responded convincingly to most of my comments and made a great effort to enhance the manuscript.*” Even more importantly, note that Referee 2, mentioned in the report of Referee 1, has already fully approved our publication without asking for further changes.

However, below, and in the revised text, please find our answer to the remaining issues raised by this referee.

New Comment 1: Comment 2 of Referee 1: I think that the new passages in the text are a valuable addition. The point of this comment, which also relates to Comment 2 of Referee 2, is to assess the reliability of the methodology across different materials. I can understand the arguments put forward by the referees. Nevertheless, additional results, maybe on λ instead of T_c to provide trends, would have been insightful – particularly for a journal with broader readership.

Response: We thank this referee for this interesting comment. As discussed in the last round, there we explained why we only provide *qualitative* ideas for comparing LNO vs. cuprate superconductors. The materials are simply very different and clearly s_{\pm} is the dominant LNO channel, while d is the dominant cuprate channel.

However, because Ref. 1 still requests us to compare the λ between LNO vs. cuprate superconductors, in the revised version we have introduced (page 8) the previously gathered, but unpublished, results for λ of the bilayer $\text{Bi}_2\text{Sr}_2\text{CaCu}_2\text{O}_8$ (BSCCO). To get the same λ strength for the single-band model for BSCCO at a filling of $n = 0.85$, we need a 25% larger U , which seems to be reasonable because cuprates are widely believed to have stronger electronic correlations than nickelates. To avoid misleading the readers, we also added sentences to note that our results only provide the trend between LNO and BSCCO. To provide a more quantitative comparison in T_c between LNO and a typical cuprate, many additional complex effects must be incorporated, and more orbitals are needed in the model calculations, as well as a different cut-off frequency prefactor ω_0 if the RPA formalism is still being used.

Accordingly, in the revised version we have added sentences in blue addressing this request raised by the first referee.

New Comment 2: Comment 3 of Referee 1: The authors rationalized their choice of U . I think that it is important to stress in the paper that variations of U may result in qualitatively different results. A good paper should investigate the robustness of the key results as a function of U . I think that this is particularly relevant here since, as Referee 2 highlighted, it is not clear whether these materials are really superconducting or not.

Response: We thank the referee for this comment. As discussed in the last round, we explained the choice of U in our study and why this choice is reasonable and solid. However, we understand that checking for the stability of the results varying U is a reasonable request.

In the revised version, we have included a table of values of λ for both the s_{\pm} and d -wave channels, varying the Hubbard U in the range where RPA is stable for a pairing state, namely in the range where a spin-density-wave still did *not* become the ground state. This requires the constraint that U be in the range at or below $U = 1.05$ (please see text in blue in the revised version and for a citation to a recent preprint of ours containing similar information). In this range of U , we have confirmed that our results are stable, and s_{\pm} dominates in the full range. We have added a table (see page 7), for completeness and for the benefit of the readers.

We also discussed in the revised version that in principle RPA could be carried out using Feynman diagrams corresponding to the ordered state, namely the spin-density-wave state in this case. This is a highly nontrivial task that we are planning to present in future work. It requires the development of an entirely novel many-body methodology, with dressed propagators and other technicalities.

Regarding the comment of the referee on whether the material is truly superconducting, to our best knowledge follow-up experimental papers already reproduced the original results by Meng Wang's group with regards to resistivity. About the missing Meissner effect, the hypothesis that is starting to dominate in arXiv papers relies on the possible "filamentary" nature of the superconducting state in this compound. Namely, only a fraction of the sample is truly superconducting. More experimental work is needed to confirm this hypothesis, and our theoretical work relies on assuming that the material is indeed superconducting, as currently claimed by the experimental groups.

In partial summary, within the limitations of the RPA many-body technique that we employed, we confirmed that s_{\pm} is the most dominant channel for pairing after varying U , as requested.

In our opinion, we have properly answered all the comments and requests of the first referee and our paper should now be ready for publication.

Response to the Second Referee

General comment: The authors have appropriately responded to all the comments and criticisms of the reviewers. I think that the paper is now suitable for publication in Nature Communications.

Response: We thank the second referee for his/her careful reading and recommendation for the publication of our work.

Response to the Third Referee

We thank the third referee for his/her careful reading and valuable comments. This referee writes “*We would like to thank the authors for their work in answering our questions and improving the manuscript. We are just left with a couple of points on which we would like further clarification on their end.*” Below, and in the revised text, we have addressed all the questions/comments of this referee.

Comment 1: Now that the authors have stated that the energy curves are obtained within FLL, we would like for them to clarify why the results here are different from those shown in their previous work. In Phys. Rev. B 108, L180510 (2023) the order (from lower to higher energy) at the same U and J values shown here is: stripe, A-type AFM, G-type AFM. Here, at the same value of U and J the order changes to stripe, G-type AFM, A-type AFM. Granted that the stripe is still the ground state but for reproducibility, the authors should clarify where the difference between the two sets of calculations is coming from.

Response: Thanks for the comment. Perhaps there was a confusion in reading the results in our Phys. Rev. B 108, L180510 (2023) paper. We fully understand this potential confusion, because all plots of energies of the many competing states are performed at a fixed pressure, which may change from plot to plot, and at a fixed ratio J/U , which also may change from plot to plot. The comment of this referee simply reminded us that we must be more careful in quoting with clarity at what fixed pressure and Hund coupling we are working. So, we thank the referee for bringing this apparent inconsistency to our attention.

In the PRB L paper, Figure 4, reproduced below with caption included, we have worked at a fixed value of pressure 29.5 GPa, and at a fixed $J/U = 0.2$, varying U , as explained in the PRB L text. The relative order of the competing phases depends on the specific value of U .

FIG. 4. (a) Calculated energies and (b) magnetic moment of different magnetic configurations vs U , in the HP phase (29.5 GPa) of LNO [67]. The G -type Néel AFM (G-AFM) configuration is taken as the energy reference. Inset in (a): Zoom of the energy difference A-type AFM (A-AFM) vs stripe, from $U = 4.9$ to 5.5 eV, as well as (b) A-AFM vs FM, from $U = 6$ to 6.5 eV.

From the PRB L figure reproduced above, panel (a), we can see that at $U = 4$ eV and $J = 0.8$ eV, the order (from lower to higher energy) is **stripe**, **G-AFM**, **A-AFM** (slightly higher than G-AFM), and then **FM**.

Let us move now to the paper under consideration in Nat Comm here. In Figure 4b reproduced below, we present the magnetic phase diagram varying the pressure P at a fixed value of U . Note that this is at $U=4$ eV and $J=0.8$ eV, with $J/U=0.2$, exactly the same as discussed in the previous paragraph regarding the figure of the PRB Letters above. At the pressure of 29.5 GPa, the order is the same: stripe, G-AFM, A-AFM (and FM is not shown because it is outside the energy range). We added a paragraph on page 4 clarifying this matter.

Note that while the order of phases is exactly the same, the energy differences are *not* which perhaps originated the comment of Ref. 3. The reason is that at 29.5 GPa (PRB L paper) we

relaxed the atomic positions and used the experimental lattice constants at 300 K provided by the original experimental Nature paper. However, in our Nat Comm under discussion, we optimized the atomic positions and lattice constants at 0 K. This leads to energy differences between various magnetic states, but the relative order is not changed.

We wish to remark that the essential physics is the same in those two sets of calculations. Specifically, the antiferromagnetic (AFM) coupling is related to the intraorbital hopping mechanism ($\sim t^2/U$), while the ferromagnetic (FM) coupling appears to be caused by interorbital hopping via the help of the Hund coupling (please see our previous work PRL127-077204-2021 where we present a mechanism to generate FM tendencies in second-order perturbation theory around the atomic limit). We will elaborate further on this topic in future publications.

Following the advice of the referee, and to help the readers, we have added a couple of sentences in the revised text (page 4) to express the compatibility between the published PRB Letters results and the present results considered for Nat Comm. The small differences arise from the different treatment of the lattice. The value of U and J are the same in both cases.

Comment 2: Concerning the presence of the vHS at the Fermi level, toning down the statements related to 1) the nesting and 2) the pairing might not be sufficient. On 1) If one were to shift the Fermi energy far from the van Hove singularity (in the DFT calculations is ~ 0.2 eV above it) the susceptibility will for sure change and whether the peaks hinting a nesting for CDW formation remain is not clear (the authors do get a stripe as the ground state and maybe it is best to just say that if the vHS turns out to be crucial for the nesting features in the susceptibility). On 2) perhaps such a shift for the pairing will remove the 'artifact' of lambda for pairing at ambient pressure and hence give rise to more realistic results.

Response: These are also good suggestions by this referee. Yes, we agree that the peak in the spin susceptibility of Fig. 5 at ambient pressure will be much reduced if the vHs shifts further down from the Fermi level. Thus, the vHS is indeed crucial for the sharp peak features in the spin susceptibility. In the revised version, we have added proper sentences to the main text (page 6, in blue) to properly alert the readers of this fact. Moreover, in Figure 7, we have now *removed* the result at $\lambda=0$ and we start at 5 GPa.

In fact, based on our very recent efforts (unpublished), we are becoming pessimistic about the possibility of stabilizing the Fmmm phase at 0 GPa to obtain superconductivity. Thus, we

have no problem in accepting the suggestion of this referee since indeed ambient pressure and the Fmmm phase appear, thus far, incompatible with one another.

In our opinion, we have properly answered all the comments and requests of the third referee and incorporated many useful sentences into the revised version, and our paper should now be ready for publication.

Response to the Fourth Referee

We thank the fourth referee for his/her careful reading, valuable comments, and recommendations. This referee writes "*I co-reviewed this manuscript with another reviewer. My comments have been incorporated into this report.*"

=====

Summary of the main changes

1. All the main changes introduced in the main text have been marked in blue.
2. Refs. [59] and [62] were added.